# OceanSODA-ETHZ: A global gridded data set of the surface ocean carbonate system for seasonal to decadal studies of ocean acidification

Luke Gregor[1] and Nicolas Gruber[1]

[1]Environmental Physics, Institute of Biogeochemistry and Pollutant Dynamics, ETH Zurich, 8092 Zürich, Switzerland

**Correspondence:** Luke Gregor (luke.gregor@usys.ethz.ch)

**Abstract.** Ocean acidification has profoundly altered the ocean's carbonate chemistry since preindustrial times, with potentially serious consequences for marine life. Yet, no long-term, global observation-based data set exists that allows to study changes in ocean acidification for all carbonate system parameters over the last few decades. Here, we fill this gap and present a methodologically-consistent global data set of all relevant surface ocean parameters, i.e., dissolved inorganic carbon ($DIC$),

total alkalinity (TA), partial pressure of $CO_2$ ($pCO_2$), pH, and the saturation state with respect to mineral $CaCO_3$ ($\Omega$) at monthly resolution over the period 1985 through 2018 at a spatial resolution of 1x1°. This data set, named OceanSODA-ETHZ, was created by extrapolating in time and space the surface ocean observations of $pCO_2$ (from the Surface Ocean $CO_2$ ATlas (SOCAT)) and total alkalinity (TA, from the Global Ocean Data Analysis Project (GLODAP)) using the newly developed Geospatial Random Cluster Ensemble Regression (GRaCER) method. This method is based on a two-step (cluster-regression) approach, but extends it by considering an ensemble of such cluster-regressions, leading to improved robustness. Surface ocean DIC, pH, and $\Omega$ were then computed from the globally mapped $pCO_2$ and TA using the thermodynamic equations of the carbonate system. For the open ocean, the cluster regression method estimates $pCO_2$ and TA with global near-zero biases and root mean squared errors of 12 $\mu$atm and 13 $\mu$mol kg$^{-1}$, respectively. Taking into account also the measurement and representation errors, the total uncertainty increases to 14 $\mu$atm and 21 $\mu$mol kg$^{-1}$, respectively. We assess the fidelity of the computed parameters by comparing them to direct observations from GLODAP, finding surface ocean pH and $DIC$ global biases of near zero, and root mean squared errors of 0.023 and 16 $\mu$mol kg$^{-1}$, respectively. These uncertainties are very comparable to those expected by propagating the total uncertainty from $pCO_2$ and TA through the thermodynamic computations, indicating a robust and conservative assessment of the uncertainties. We illustrate the potential of this new dataset by analyzing the climatological mean seasonal cycles of the different parameters of the surface ocean carbonate system, highlighting their commonalities and differences. Further, this dataset provides a novel constraint on the global and basin-scale trends in ocean acidification for all parameters. Concretely, we find for the period 1990 through 2018 global mean trends of 8.6 $\pm$ 0.1 $\mu$mol kg$^{-1}$ decade$^{-1}$ for $DIC$, $-0.016 \pm 0.000$ decade$^{-1}$ for pH, 16.5 $\pm$ 0.1 $\mu$atm decade$^{-1}$ for pCO$_2$, and -0.07 $\pm$ 0.00 decade$^{-1}$ for $\Omega$. The OceanSODA-ETHZ data can be downloaded from https://doi.org/10.25921/m5wx-ja34 (Gregor and Gruber, 2020).

# 1 Introduction

The oceans have taken up roughly one quarter of the anthropogenic $CO_2$ that has been released into the atmosphere since the start of the industrial era (Sabine et al., 2004; Gruber et al., 2019), modulating the increase in atmospheric $CO_2$ substantially. However, this buffering of anthropogenic climate change by the ocean comes with a substantial cost, i.e., ocean acidification (Doney et al., 2009). The uptake of anthropogenic $CO_2$ over the last 150 years has made the surface ocean more acidic with a decrease in global mean pH from ~8.2 around 1850 to ~8.1 today (Feely et al., 2009; Jiang et al., 2019). This decrease in pH

equates to a ~30% increase in the concentration of $H^+$ ions. Some of the anthropogenic $CO_2$ taken up from the atmosphere remains in the seawater as dissolved $CO_2$, thus increasing its partial pressure ($pCO_2$). In fact, surface ocean $pCO_2$ tends to track the increase in atmospheric $pCO_2$ rather closely (e.g. Bates et al. 2014) owing to the $\sim$ 1-year timescale for the equilibration of $CO_2$ across the air-sea interface (Sarmiento and Gruber, 2006), which is smaller than the decadal timescale increase in atmospheric $CO_2$ (Friedlingstein et al., 2019). While some of the added $CO_2$ stays as $CO_2$, the majority of is titrated away by

the ocean's carbonate ion (Sarmiento and Gruber, 2006), leading to a substantial reduction in its concentration. This reduces the saturation state ($\Omega$) with regard to the mineral calcium carbonate ($CaCO_3$), where an $\Omega$ of < 1 leads to dissolution of $CaCO_3$.

These chemical changes, collectively described as ocean acidification, will have a profound impact on marine organisms, especially those that form shells made of $CaCO_3$ (Orr et al., 2005; Fabry et al., 2008; Doney et al., 2009; Bednaršek et al., 2019; Doney et al., 2020). Calcifying organisms living in high latitudes and subtropical and tropical upwelling regions, with

their naturally low $\Omega$ and pH, may be particularly vulnerable, as these regions will be among the first to cross critical saturation thresholds (Orr et al., 2005; Steinacher et al., 2009; Gruber et al., 2012; Franco et al., 2018; Fabry et al., 2009; Hauri et al., 2016; Negrete-García et al., 2019). However, marine organisms may be susceptible to changes even where $\Omega > 1$ due to a shift in energetic requirements for shell formation (Orr et al., 2005; Pörtner and Farrell, 2008). For example, it is well known that corals start to decrease their calcification already at saturation states well above 3 (Gattuso et al., 1998). Ocean acidification

will thus have a significant economic impact on fisheries and tourism through the impact on shellfish and corals, respectively (Cooley and Doney, 2009; Doney et al., 2020).

At the global scale, most of what we know about the progression of ocean acidification in the recent decades has come from either models (Bopp et al., 2013; Kwiatkowski et al., 2020) or from the combination of model-based trends with observation-based climatologies (Feely et al., 2009; Jiang et al., 2019). A notable exception are the large number of studies that have

analyzed the trends and variability of surface ocean $pCO_2$ (e.g. Landschützer et al. 2013, 2016; Rödenbeck et al. 2014; Denvil-Sommer et al. 2019; Gregor et al. 2019) and the effort of Lauvset et al. (2015) and Turk et al. (2017) to analyze long-term trends in pH and $\Omega$ respectively. But these studies remained limited to one single parameter. At the local-to-regional scale, a number of long-term timeseries have provided excellent insights into the processes and trends of ocean acidification across all carbonate system parameters (e.g. Bates et al. 2014), but no global comprehensive view of the historical development of ocean acidifi-

cation based on observations exist. This is largely a consequence of the limited observations, although observational efforts have increased substantially in the recent decades through efforts such as GOA-ON (Global Ocean Acidification Observing Network (Tilbrook et al., 2019). The OceanSODA (Satellite Oceanographic Datasets for Acidification) project (https://esa-

oceansoda.org), which this study forms part of, aims to close this gap by linking satellite observations with in situ observations of the marine carbonate system.

In line with the goal of the OceanSODA project, we aim to develop a global, observation-based data set documenting the progression of ocean acidification over the recent decades. Such a data set will be crucial to put the current trends of ocean acidification into the context of the changes over the last few decades. By also describing the level of variability in ocean acidification around the long-term trend, it will also help to better understand the challenges that marine organisms are facing. Additionally, it will permit us to explore in much more detail how ocean acidification has unfolded regionally, and potentially

deviated from the simple model of it being dependent on the rise in atmospheric $CO_2$.

    The well-measurable parameters of the marine carbonate system are dissolved inorganic carbon (DIC), total alkalinity (TA), pH, and the partial pressure of carbon dioxide ($pCO_2$). Very few measurement programs measure all of these parameters concurrently. In fact, the vast majority of the observational programs measure only one parameter, with $pCO_2$ being the most often measured one, followed by DIC, TA, and pH (Bakker et al., 2016; Olsen et al., 2019). Since two parameters are sufficient to

fully describe the marine carbonate system, any combination of two will permit to fully reconstruct the entire carbonate system. But not all combination are equally suited, given the uncertainties in the measurements, the uncertainties in the coefficients of the carbonate chemistry, and the spatio-temporal coverage vis-à-vis the variability of these parameters.

    We use here the pair $pCO_2$ and TA as the basis for our reconstruction for two reasons. First, these are the best observed parameters relative to their spatio-temporal variability, permitting us to develop better predictive models for the global surface

ocean than possible for, e.g., $DIC$ and pH. Second, detailed assessments of the internal consistency of the oceanic carbonate system have shown that $pCO_2$ and TA are a well suited pair to estimate pH, owing to the reliability of the measurements and the predictive accuracy (Bockmon and Dickson, 2015; Bakker et al., 2016; Raimondi et al., 2019). This is not the case if $DIC$ was used instead of TA. Our choice is supported by Takahashi and Sutherland (2013), who developed the first seasonal climatology of all surface ocean carbonate system parameters using the same pair.

Measurements of $pCO_2$ are abundant compared to the other variables, due to a well-established and robust underway sampling protocol that allows instruments to also be installed on non-scientific vessels under the Volunteer Observing Ship (VOS) program (Bakker et al., 2016; Pierrot et al., 2009). High quality $pCO_2$ data are also easily accessible thanks to SOCAT that consolidates underway $pCO_2$ observations and ensures the quality of observations (Bakker et al., 2016). Total alkalinity is not as widely measured as $pCO_2$ due to the fact that measurements are made discretely with bottle samples (Dickson et al., 2007).

But, fortunately, TA is highly correlated with salinity on a global scale (r=0.96) making it a suitable variable for prediction with a < 10% error of the observed range (Lee et al., 2006; Olsen et al., 2019; Broullón et al., 2018). Further, the accessibility of TA measurements is made possible through the continued efforts of GLODAP (Olsen et al., 2019). We discarded the option to use $DIC$ instead of TA, even though $DIC$ is slightly more often sampled than TA. This decision is based on the fact that $DIC$ is more variable than TA, and also its correlation with salinity is much lower. As a result, it is difficult to develop predictive

models for $DIC$ that are as accurate and precise as those for TA. Oceanic pH is also not an option, since historically it has been measured far less often than the other parameters. This is changing, since progress with reference materials and new sensors

have permitted a tremendous increase in the number of pH measurements in recent years, largely benefiting from deployments of biogeochemical Argo floats (Claustre et al., 2020).

The actual spatial and temporal coverage for any of these parameters is very low. Even for $pCO_2$, i.e., the parameter with the densest coverage, only about 1.4% of the global surface ocean has been sampled in any given month over the past 30 years (Bakker et al., 2016). Thus, the global-scale reconstruction of the progression of ocean acidification requires a very substantial inter- and extrapolation effort. Advances in remote sensing (Land et al., 2019), and the increasing power and usability of machine learning techniques have permitted us to address this challenge, leading to a proliferation of such efforts. However, they vary greatly between the different parameters of the marine carbonate system.

By far the most established efforts are those that interpolate and extrapolate the ocean $pCO_2$ observations, as demonstrated by the inter-comparison project by Rödenbeck et al. (2015). Feed-forward neural networks (FFNN) have become one of the favored tools (Landschützer et al., 2013; Zeng et al., 2014; Denvil-Sommer et al., 2019), but other statistical and machine learning methods, such as Bayesian regression and tree-based regression, have also been used with similar success (Rödenbeck et al., 2014; Gregor et al., 2019). However, the specific implementation of the methods is what sets the assortment of methods apart. For example, the SOM-FFN method of Landschützer et al. (2013) and the CSIR-ML6 method of Gregor et al. (2019) (amongst others) first cluster the data based on a certain set of climatological predictors and then perform a regression on $pCO_2$ for each resulting cluster. An alternate approach, used by both the CMEMS-FFNNv2 (Denvil-Sommer et al., 2019) and NIES-FNN (Zeng et al., 2014) methods, is to include the positional coordinates, without the need for subsetting the data by clustering. Despite the differences in implementation and regression algorithms, the majority of methods achieve for the open ocean a RMSE of roughly 18 $\mu$atm when compared with SOCAT (Gregor et al., 2019). However, each of these methods has its strengths and weaknesses; for example, the SOM-FFN and CSIR-ML6 methods are able to generalize estimates in data-sparse regions due to information sharing within a cluster, but the methods suffer from discrete boundaries where clusters meet (Gregor et al., 2019). These discrete boundaries may introduce artefacts when applied to certain questions. This is also the case, for example in the blended open ocean - coastal ocean product of Landschützer et al. (2020), where the authors combined the open ocean estimate of Landschützer et al. (2016) with the coastal product of Laruelle et al. (2017).

The extrapolation of TA onto a global grid is also well established (Gruber et al., 1996; Millero et al., 1998; Lee et al., 2006; Takahashi and Sutherland, 2013; Good et al., 2013; Carton et al., 2018; Bittig et al., 2018). The highly-linear relationship between salinity and TA means that linear regressions have been able for quite some time to estimate TA with adequate accuracy. For example, Gruber et al. (1996) developed a globally-applicable multi-linear regression model involving salinity and the conservative tracer PO (PO = $O_2$+170·$PO_4$, (Broecker and Peng, 1974)) and achieved a global RMSE of 11 $\mu$mol kg$^{-1}$. Lee et al. (2006) also used a MLR approach, but differentiated it regionally using salinity, temperature and spatial coordinates as independent variables. The same approach was followed by Takahashi et al. (1993). More recently, more nuanced and non-linear regression approaches have improved upon the MLR approaches (Sasse et al., 2013; Carter et al., 2018; Broullón et al., 2018; Bittig et al., 2018). For example, the Locally Interpolated Alkalinity Regression (LIARv2) still makes use of linear regression but interpolates the regression coefficients spatially from a fixed set of trained regression nodes located at every 5[th] point (Carter et al., 2016). Sasse et al. (2013) used a self-organizing map approach coupled with a local linear optimizer (called

SOMLO) and achieved a global RMSE of 9 $\mu$mol kg$^{-1}$. A similar RMSE was achieved by Broullón et al. (2018) using a neural network approach (NNGv2). These low RMSE levels were also achieved by these studies avoiding the nearshore and coastal environments, where variability in the surface ocean carbonate system is much higher than in the open ocean (Laruelle et al., 2017). In addition to these global regressions, several regionally-specific regressions were developed (see Table 1 in Land et al. 2019).

In comparison, very few efforts attempted to inter- and extrapolate DIC. Lee et al. (2000) were the first to produce a global map of $DIC$ using a regression methodology; however, their application employed a regional multi-linear regression model similar to that used later to map TA. But their application was limited to the generation of a seasonal climatology. It was not until Sasse et al. (2013) when the first global reconstruction of the temporal progression of $DIC$ over multiple years was published. They used the same SOMLO method as they had used for TA, creating global maps of $DIC$ with a RMSE of 11 $\mu$mol kg$^{-1}$. More recently, Keppler et al. (2020) used the SOM-FFN method of Landschützer et al. (2013) to reconstruct $DIC$ throughout the upper water column on a monthly basis, but they limited their discussion to the mean seasonal cycle.

Here, to map TA and pCO$_2$ globally, including the coastal ocean, the Arctic, and the Mediterranean, we use a newly-developed two-step cluster-regression approach that is similar in design to the SOM FFN method (Landschützer et al., 2013, 2016) but extend it by using an ensemble of such cluster-regressions. This method, referred to as Geospatial Random Cluster Ensemble Regression (GRaCER), increases the robustness of the estimates considerably. It also removes the boundary problems inherent in all methods that use fixed regional boundaries. We apply the same methodology to TA and pCO$_2$, resulting in methodologically-consistent global estimates of the two parameters, from which DIC, pH, and $\Omega$ can then be computed using the well-established thermodynamic models of the seawater carbonate system. These latter estimates can then be compared against the many available $DIC$ and pH measurements, providing a large set of independent data to assess the fidelity of our estimates. This requires also a good understanding of the different sources of uncertainties, including those emanating from sampling and measurement, from the statistical modeling, and from the lack of representativeness, i.e., the fact that a local measurement is not representative for the large pixel (100 x 100 km and 1 month), that one models in our regressions.

The rest of the manuscript describes the data and methods used to calculate this data set for ocean acidification. The uncertainties of the predictions are assessed, followed by the presentation of the data with a focus on the seasonal cycle. Last, we discuss the implications of the uncertainties for the use of the derived marine carbonate system.

## 2 Methods

To reconstruct the global progression of all parameters of the surface ocean carbonate system over the last three decades (1985 through 2018), we follow the three steps depicted by the flow diagram in Figure 1. First, we develop a statistical model for the measured TA and pCO$_2$ using the newly developed GRaCER method. This method itself consists of two steps, i.e., a cluster step, where the target variables are clustered regionally, and a regression step, where for each cluster, a regression is evaluated. These two steps are repeated multiple times, creating an ensemble of models; Second we map these two quantities globally and over time using this ensemble of statistical models and global observations of the predictor variables; Third and last, we use a

thermodynamic model of the seawater carbonate system to compute the remaining parameters of the surface ocean carbonate system, namely $DIC$, pH, and $\Omega$. Along the way, we extensively evaluate and test each step with independent observations. We refer to the dataset with the evaluated and complete marine carbonate system as OceanSODA-ETHZ.

Next, we describe the concept of the GRaCER method, and then detail its implementation for $p\text{CO}_2$ and TA. This is followed by a description of the numerous types of data employed and how they were prepared. Lastly, we demonstrate how we used a thermodynamic model to derive the remaining parameters of the marine carbonate system.

## 2.1 GRaCER Algorithm

The GRaCER algorithm builds conceptually on a series of cluster-regression algorithms that have been successfully used for the inter- and extrapolation of surface ocean $p\text{CO}_2$ (Sasse et al., 2013; Landschützer et al., 2013, 2016; Iida et al., 2015; Gregor et al., 2019). The main advantage of such a two-step approach is that the first clustering step organizes the variability regionally and temporally. This greatly enhances then the fidelity of the second step, i.e., the regression, as the size of the regression problem is reduced from the global domain to smaller, more homogeneous regions. A second advantage is that this clustering brings together regions with similar seasonality and similar co-variability with potential predictors, irrespective of the number of observations. The regression step explains the variability within each region over time and space dimensions, including interannual variability. Further, the clustering permits the regression to transfer information from spatially distant, but geochemically similar regions, making the inter and extrapolation more robust in data poor regions. The main innovation of the GRaCER algorithm relative to the previously used two-step approaches is its use of ensembles of cluster-regressions, i.e., the generation of a whole series of clusters and corresponding regressions, which overcomes the boundary problems that are inherent in all two-step approaches.

For the **clustering** step, we use monthly climatological data of $p\text{CO}_2$ and TA and related parameters (Figure 2a-c), to determine the main patterns of variability of the target variable and its co-variability with potential predictor variables. We opted for a clustering on climatological data rather than on the actual monthly data in order to clearly focus on the clustering step's role to isolate primarily regions with the same seasonal cycle. The alternative approach, i.e., to cluster on the monthly data is also more prone to errors since the climatological distributions are better known than their month-to-months variations. Finally, clustering on monthly data would also take away signals from the regression step, which is actually better suited to capture the smaller level variations associated with the interannual variability and trends. The mini-batch K-means implementation in the Python *Scikit-Learn* package is used to perform the clustering due to its computational efficiency and scalability with large data sets (Pedregosa et al., 2011). A user-defined number of cluster centers are initiated, where cluster centers represent the mean of the points in a cluster. The K-means++ algorithm is used to initiate the cluster centers, which randomly selects the location of the first cluster center, then iteratively selects a best-guess location for the remaining cluster centers. Thereafter, the algorithm minimizes the distance between cluster centers and data points in the variable space. Once the clusters have been defined for the climatological domain, the co-located training data are assigned to the monthly clusters.

The **Regression** is then performed individually for each of the clusters (Figure 2d-f). The GRaCER method does not use a prescribed regression method — rather the appropriate algorithm for the particular use case is implemented. Importantly, the

algorithm must be able to scale appropriately to the size of the problem. For example, the training data set for TA is one $20^{th}$ of the size of the $p$CO$_2$ training data set, thus a more computationally-expensive method can be used to predict TA.

The **Ensemble** members are created by performing the cluster-regression step multiple times. Creating an ensemble is possible due to the fact that each clustering instance is slightly different (Figure 2g-i). In practice, the spatial distribution of the clusters is similar, *i.e.,* there is consistency in the typology of the clusters, particularly in regions where clusters are well defined, such as in the subtropical gyres and in the tropical eastern Pacific. However, there are regions that belong to different clusters, *i.e.,* there is slight variance in the typology between ensemble members. The differences are due to the random initialization of the first cluster center in the K-means clustering step, and the fact that clustering variables for some regions have weak gradients in spatial auto-correlation resulting in weak association to a cluster. In practice, this means that the location of cluster boundaries vary between ensemble members, thus the ensemble mean does not have discrete boundaries (Figure 2j).

## 2.2 Algorithm Implementation

### 2.2.1 Total alkalinity

For the estimation of TA, we employ the support vector regression (SVR) regression method with 12 clusters and 16 ensemble members. The clustering is performed on climatological mean TA, sea-surface salinity (SSS), sea-surface temperature (SST) and nitrate (NO$_3^-$; Table 1 and section 2.3 below). The optimal variables on which clustering should be performed were selected by assessing the regression scores of each combination of variables following the methodology of Gregor et al. (2019). All data are standardized to the mean ($\mu$) and standard deviation ($\sigma$) prior to clustering ($\frac{(x-\mu)}{\sigma}$), after which TA is given three times the weight of the other variables.

A similar exhaustive search was used for determining the number of clusters. The number of ensemble members was chosen by the number above which there is no longer an increase in performance, analogous to the number of trees in a Random Forest. Test data are a subset of years spaced three years apart starting in 1985. We ensure that the models are not overfitted by selecting hyper-parameters using K-fold cross validation (further details are in Section A3).

To regress and map TA, we use SSS, SST, silicic acid ($Si$), and N$^* =$ NO$_3^- - 16 \cdot$ PO$_4^{3-}$ (simplified from Gruber and Sarmiento, 1997) as predictors. Our choice of SSS and SST as predictors is easily justified by these two variables accounting for the majority of TA variability (Lee et al., 2006; Carter et al., 2018). The addition of $Si$ and N$^*$ as predictors is to account for seasonal changes in primary production that has an impact on TA (Wolf-Gladrow et al., 2007; Carter et al., 2018). Further, N$^*$ expresses the zonal differences between and within the large ocean basins better than using simply NO$_3^-$ or PO$_4^{3-}$ — an important consideration, since coordinates (i.e., latitude and longitude) are not included in our set of predictors.

### 2.2.2 Partial pressure of CO$_2$

For the estimation of $p$CO$_2$, we use two regression methods, i.e., GBDT (gradient boosted decision trees) and FFNNv2 (feed forward neural network). These are implemented with 21 clusters and 16 ensemble members (eight each). The number of clusters is at the upper end of the range compared with the number of clusters used by the MPI-SOMFFN or CSIR-ML6

methods. However, testing has shown that additional clusters are required to account for the additional complexity by our inclusion of data from the coastal, Arctic, and Mediterranean seas.

Clustering is performed on climatological values of $pCO_2$, SST, mixed layer depth, and Chlorophyll-$a$, with additional weighting given to $pCO_2$. As with TA, all variables are standardized prior to clustering with $\frac{(x-\mu)}{\sigma}$, after which $pCO_2$ is multiplied by 3 to give it stronger weighting.

Details of the regression method, and of the hyper-parameter selection are given in section A3. Test data are selected as every $5^{th}$ year starting in 1985, and validation data for early stopping is selected using the same approach starting in 1987, where the latter is used for early stopping to reduce over-fitting and keep model complexity within bounds.

The regression and mapping is performed with the following variables as predictors: SST, SSS, the logarithm of Chlorophyll-$a$, the logarithm of mixed-layer depth, the meridional and zonal components of the surface winds, the sine and cosine of $\frac{\text{day of year} \cdot \pi}{365 \cdot 180}$, and the atmospheric dry-air mixing ratio ($x$CO$_2$). These predictors are the same as used by Gregor et al. (2019) and various combinations of these methods have been used by previous approaches (Landschützer et al., 2014; Denvil-Sommer et al., 2019).

It is important to note that the predictors are proxies for the spatio-temporal changes in $pCO_2$ and do not necessarily explain the physical mechanism by which changes in $pCO_2$ are driven. For example, an increase in sea-surface temperature in the subtropics results in an increase $pCO_2$ as shown by Takahashi et al. (1993); Lefèvre and Taylor (2002). In contrast, surface warming in the Southern Ocean can be a proxy for stratification that reduces outcropping of high CO$_2$ waters (Landschützer et al., 2015; Gregor et al., 2018). Similarly, changes in SSS and MLD also capture the distribution and processes that drive changes in surface $pCO_2$, such as stratification and mixing. However, the climatological MLD product used here does not capture interannual variability in stratification and mixing. We thus include the two surface wind components that, along with SST, are a proxy for wind-driven mixing and upwelling. Chlorophyll is also an important driver of $pCO_2$ on a local scale, particularly in the high latitude regions where high primary productivity results in rapid uptake of $pCO_2$ (Bakker et al., 2008; Gregor et al., 2018). Lastly, $x$CO$_2$ is included to account for the close tracking of oceanic $pCO_2$ to atmospheric CO2 concentrations (Bates et al., 2014).

## 2.3 Data

Data are used to develop the two-step GRaCER model, i.e., clustering and regression, and to evaluate the estimates. Table 1 provides an overview of all data employed and the purposes for which they are used and Table 2 shows the corresponding source of the data. We describe each data set by parameter and use.

### 2.3.1 Data for clustering

For the clustering of TA, we used the mapped product of total alkalinity (TA$_{\text{map}}$) from the GLODAPv2 (Lauvset et al., 2016). This product represents a quasi-annual mean as it was generated without consideration of the seasonal cycle. We repeat this quasi-annual mean TA to create a monthly data set over which clustering can be performed. We thus assume that the spatial variability of TA is larger than the seasonal variability. This is backed by Takahashi and Sutherland (2013) and Broullón et al.

**Table 1.** Variables used as the clustering features and predictor variables for regression. Details about these data are given in the text. Note that clustering features are all resampled to monthly climatologies. $^{†}$TA$_{map}$ is the exception where a quasi-annual mean is used. The first column for regression shows the target variable. All machine learning models use the same variables to train and predict the final estimates with the exception of $^{‡}$SSS, where ungridded GLODAP data is used to train and SODA salinity is used to predict. All other references to SSS refer to the gridded SODA product.

| Clustering | Clustering features (monthly climatology) |
| --- | --- |
| TA | $^{†}$TA$^{map}$, SSS, SST, $N^*$ |
| $p$CO$_2$ | $p$CO$_2^{map}$, SST, Chl-a, MLD$_{clim}$ |
| **Regression & Mapping** | Predictors (monthly) |
| TA$^{GLODAP}$ | $^{‡}$SSS$_{SODA}^{GLODAP}$, SST, $Si$, $N^*$ |
| $p$CO$_2^{SOCAT}$ | $x$CO$_2^{atm}$, SST, SSS, Chl-a, MLD$_{clim}$, $u$-wind, $v$-wind |

**Table 2.** Data sources used in this study. *C3S (2017) is short for Copernicus Climate Change Service (C3S) (2017).

| Product name | Variable | Abbrev | Reference |
| --- | --- | --- | --- |
| SOCAT | $p$CO$_2^{SOCAT}$ | $p$CO$_2$ | Bakker et al. (2016) |
| MPI-SOMFFN | Gap-filled $p$CO$_2$ | $p$CO$_2^{map}$ | Landschützer et al. (2016) |
| Jena-MLS | | | Rödenbeck et al. (2014) |
| LSCE-FFNN | | | Denvil-Sommer et al. (2019) |
| LDEO | | | Takahashi et al. (2014) |
| GLODAP | Total Alkalinity (in-situ) | TA | Olsen et al. (2016) |
| | Total Alkalinity (mapped) | TA$^{map}$ | Lauvset et al. (2016) |
| OSTIA | Sea surface temperature | SST | Good et al. (2020) |
| | Sea-ice fraction | ICE | |
| SODA v3.4.2 | Salinity | SSS | Carton et al. (2018) |
| | Mixed Layer Depth | MLD | Holte et al. (2017) |
| ERA5 | Sea-level pressure | Pres | *C3S (2017) |
| | U-component of wind | $U$ | |
| | V-component of wind | $V$ | |
| NOAA: ATM | Mole fraction of CO$_2^{atm}$ | $x$CO$_2^{atm*}$ | Dlugokencky et al. (2019) |
| Globcolour | Chlorophyll-a | Chl-a | Maritorena et al. (2010) |
| WOA | Phosphate | PO$_4^{3-}$ | Boyer et al. (2013) |
| | Nitrate | NO$_3^-$ | |
| | Silic acid | $Si$ | |

(2018) who found that the seasonal variability of TA for the majority of the ocean was more than a factor of 10 smaller than the spatial variability.

For the clustering step of $p\text{CO}_2$, we use four data-based products resampled and gridded to a monthly by $1\times1°$ resolution ($p\text{CO}_2^{\text{map}}$), namely LDEO by Takahashi et al. (2014), MPI-SOMFFN by Landschützer et al. (2016), Jena-MLS by Rödenbeck et al. (2014), and CMEMS-FFNNv2 by Denvil-Sommer et al. (2019). It may seem tautological to use other machine learning estimates, but these data are just used to create regional clusters, i.e., they are not used in the regression step. Relative to previous two-step approaches Landschützer et al. (2016); Denvil-Sommer et al. (2019), which used just the LDEO product, we expanded on this by including three more estimates. In doing so, we make the implicit assumption that this ensemble of estimates is a better representation of the $p\text{CO}_2$ monthly climatology than the LDEO climatology alone.

SSS is from the Simple Ocean Data Assimilation (SODA) analysis (Carton et al., 2018) and SST from the Operational Sea Surface Temperature and Sea Ice Analysis (OSTIA) product (Good et al., 2019, 2020). $N^*$ is calculated using monthly climatologies of $\text{NO}_3^-$ and $\text{PO}_4^{3-}$ from the World Ocean Atlas updated in 2018 (Boyer et al., 2013). We use the monthly climatology of density-based mixed-layer depth (MLD) from Holte et al. (2017) that is estimated from Argo float profiles. The MLD data product merges monthly estimates of MLD from multiple years, but is averaged into a climatology due to the paucity of data on an annual scale. A two-dimensional moving average filter is applied to the MLD to interpolate missing data and remove the noise introduced by interannual and sub-monthly variability.

Chlorophyll-a (Chl-a) is from the Globcolour project where a monthly climatology is calculated for the period from 1998 through 2018 (Maritorena et al., 2010). The missing data in the high latitudes during winter are filled with a 0.3 mg/m$^{-3}$, which is roughly the $20^{th}$ percentile of global chlorophyll-a. Lastly, we take the log transformation (base 10) of Chl-a, to convert the log-like distribution of Chl-a to a normal distribution for improved performance in the gradient descent algorithm of the FFNNv2.

### 2.3.2 Data for regression and mapping

For the regression step of TA, the bottle measurements from the GLODAP v2 product are used as the target variable (Olsen et al., 2019). Following Lee et al. (2006), we select data shallower than 20 m in latitudes lower than $30°$ and shallower than 30 m at higher latitudes. We do not exclude measurements taken in nearshore and coastal environments as was previously done. The quality of TA measurements was historically not as rigorous as the SOCAT $p\text{CO}_2$ data due to the lack of reference standards before the mid-1990's (Bockmon and Dickson, 2015). However, most of the biases in the cruises were corrected based on calibration to deep samples, where it is assumed that interannual TA variability is negligible relative to the magnitude of the bias (Olsen et al., 2019). These bias corrections amount to $\pm 5$ $\mu\text{mol kg}^{-1}$ on average.

For the regression of $p\text{CO}_2$, we use SOCAT v2019 where only data with a SOCAT cruise quality flag of A to D and a WOCE quality flag of 2 are used. As is the case for TA, we do not exclude data from coastal and nearshore environments. The fugacity of $\text{CO}_2$ ($f\text{CO}_2$) reported in SOCAT v2019 is converted to $p\text{CO}_2$ using:

$$p\text{CO}_2 = f\text{CO}_2 \cdot \exp(P_{\text{atm}}^{\text{surf}} \cdot \frac{B + 2 \cdot \delta}{R \cdot T^{\text{SOCAT}}}) \tag{1}$$

where $P$ is atmospheric pressure at sea-level from the ERA5 reanalysis product (Copernicus Climate Change Service (C3S), 2017). $B$ and $\delta$ are virial coefficients, $R$ is the gas constant, and $T^{\text{SOCAT}}$ is the ship intake temperature in °C (Dickson et al.,

2007). In exploratory work for this study, we tested predicting $\Delta p\mathrm{CO}_2 = p\mathrm{CO}_2^{atm} - p\mathrm{CO}_2$ instead of just $p\mathrm{CO}_2$, but found that this did not produce credible results; for a more in depth discussion see Section A2.

The discrete measurements of $p\mathrm{CO}_2$ and TA are resampled on a monthly grid (Jan 1985 through Dec 2018) with a spatial resolution of $1 \times 1°$ to match the predictors used in the mapping step.

Finally, outliers are removed from gridded $p\mathrm{CO}_2^{\mathrm{SOCAT}}$ using the methods described in Section A1. In total, 2425 points are removed from the gridded $p\mathrm{CO}_2^{\mathrm{SOCAT}}$ using these outlier removal approaches, equivalent to 0.85% of the original gridded data.

We use sea-surface temperature from OSTIA for both TA and $p\mathrm{CO}_2$ regression (Good et al., 2020). The TA model is trained using in situ salinity from GLODAP v2 but salinity from SODA v4.3.2 is used for the mapping step (Olsen et al., 2019; Carton et al., 2018). The $N^*$ and $Si$ are the same as used in the clustering step, but are repeated for the number of years. Similarly, the mixed layer depth climatology described in section 2.3.1 is repeated for each year. We use the global mean of the mole fraction of $\mathrm{CO}_2$ for the marine boundary layer ($x\mathrm{CO}_2^{\mathrm{mbl}}$) as a predictor in the regression as the correction for water vapor pressure may otherwise introduce co-variance with other predictors (*i.e.* SST, SSS). Missing data in the monthly Globcolour chlorophyll-a product is filled with climatological data described in section 2.3.1. The meridional and zonal components of the surface winds are averaged from the hourly output from the ERA5 reanalysis (Copernicus Climate Change Service (C3S), 2017).

### 2.3.3 Evaluation variables

The machine learning estimates of TA, $p\mathrm{CO}_2$, and the computed $DIC$ and pH are evaluated against data that are not used in the training or mapping step.

For $DIC$ and pH, we use the directly measured data from GLODAP v2.2019 (Olsen et al., 2019). Bockmon and Dickson (2015) report a measurement error of $\pm 5$ $\mu$mol kg$^{-1}$ for GLODAP $DIC$ in an inter-laboratory comparison. Olsen et al. (2019) estimate the measurement error for pH to be 0.01. To be consistent with TA, we select data shallower than 20 m in latitudes lower than 30° and shallower than 30 m at higher latitudes and resampled the data to monthly by $1° \times 1°$.

Three long term time series stations are used to provide direct independent comparisons for $DIC$ and TA, namely: the Hawaii Ocean Time-series at 22.57°N, 158°W (HOT, Dore et al. 2009); the Bermuda Atlantic Time Series at 32°N, 64°W (BATS, Bates and Peters 2007); and the Irminger station in the high northern Atlantic (64.3°N, 28°W, Olafsson et al. 2010, only for $DIC$). The accuracy for these measurements is reported to be below 2 $\mu$mol kg$^{-1}$ for $DIC$ and $\sim$4 $\mu$mol kg$^{-1}$ for TA for all stations. We use the same depth constraints for the long term stations as for GLODAP, explained in the paragraph above. $p\mathrm{CO}_2$ is also calculated from $DIC$ and TA for HOT and BATS to provide an additional constraint.

Data present in the Lamont-Doherty Earth Observatory $p\mathrm{CO}_2$ data set, but not in SOCAT are used to independently compare $p\mathrm{CO}_2$ (Takahashi et al., 2019). Takahashi et al. (2019) report an error estimate of $\pm 2.5$ $\mu$atm, but it must be added that some of the data unique to LDEO may be excluded from SOCAT due to stricter quality control criteria for of the latter, thus errors for the LDEO data are expected to be larger (Bakker et al., 2016).

Finally, we include Argo float measurements of pH from the Southern Ocean Carbon and Climate Observations and Modeling project (SOCCOM) (Johnson et al., 2017; Williams et al., 2017). Johnson et al. (2016) report a mean uncertainty of

±0.019 for pH for the entire water column, though this is likely higher for the upper 30 m as the authors report lower errors for estimates below 50 m.

## 2.4 Computation of DIC, pH, and $\Omega$

The remaining parameters of the marine carbonate system, i.e., $DIC$, pH, and $\Omega$ are computed using the Python version of *CO2SYS* (Humphreys et al., 2020) originally developed by Lewis et al. (1998). In addition to $pCO_2$ and TA, *CO2SYS* requires the input of sea-surface temperature, sea-surface salinity, pressure (assumed 0 dBar at the surface), $PO_4^{3-}$, and $Si$. We use the same data sources described in section 2.3.2 and Table 1. Climatologies of $Si$ and $PO_4^{3-}$ are repeated for each year, thus assuming no interannual variability. The impact of this assumption is minimal ($\ll 2$ $\mu$mol kg$^{-1}$), as can be shown by varying 335 these nutrients over seasonal cycle. The dissociation constants by Dickson et al. (1990) for $K_{\mathrm{HSO_4}}$ and the total boron-salinity relationship by Uppström (1974) were used, as recommended by Orr et al. (2015) and Raimondi et al. (2019). For further details on the calculation and the full description of the marine carbonate system, see Dickson et al. (2007).

An important consideration in these calculations is the internal consistency of the marine carbonate system, i.e., the error due to uncertainties in the equations and coefficients that describe the marine carbonate system. Raimondi et al. (2019) pointed out 340 that the $pCO_2$-TA pair has the lowest error in the calculation of pH ($0.003 \pm 0.008$ pH units) using the dissociation constants by Mehrbach et al. (1973) as refitted by Dickson and Millero (1987). However, using the same pair and the same dissociation constants resulted in an estimate of $\Omega$ with respect to Aragonite that is very different from that computed using the DIC-TA pair. But since Raimondi et al. (2019) lacked direct measurements of $\Omega$, it remains unclear which pair is actually better for $\Omega$. We cannot resolve this here but need to acknowledge that this inconsistency adds some additional uncertainty to our computed 345 $\Omega$ values.

## 3 Uncertainty assessment

Any application of our data product requires a firm understanding of the errors and uncertainties associated with each of the reported parameters of the surface ocean carbonate system. We first discuss the errors and uncertainties associated with the statistically-modeled quantities TA and $pCO_2$, and then those with the computed parameters $DIC$, pH and $\Omega$. Then, we will 350 compare these propagated uncertainties with the uncertainty of the computed $DIC$ and pH by comparing these values with in situ observations. This provides a strong check on our ability to establish a full uncertainty budget. Here, we use the term "uncertainty" to characterize the range of values within which the true value is asserted to lie with some level of confidence. The term "error" is used in two ways. First, as a process that leads to deviations between the measurement and the true value, and second as an estimate quantifiable against a known value. For the purpose of our analysis here, we consider the training 355 data sets for TA and $pCO_2$ as such known values, i.e., we assume that these observations are unbiased. This can be justified on the basis of their having undergone extensive secondary quality control.

## 3.1 Sources of errors for TA and $pCO_2$

We identify three sources of errors that contribute to the total uncertainty for $pCO_2$ and TA, namely the uncertainty stemming from the measurement ($M$), representation ($R$) and prediction ($P$) errors. Assuming independence of the three error sources, the total uncertainty ($E$) for the TA and $pCO_2$ estimates can thus be expressed as the root of the squared sum of the uncertainties from the three error sources:

$$E = \sqrt{M^2 + R^2 + P^2} \tag{2}$$

The **measurement error** reflects the combination of potential biases (systematic errors) from sampling and measurement as well as random errors associated with sampling and the imprecise nature of the measurement system. Since both TA and $pCO_2$ are being measured against certified reference materials and have undergone extensive secondary quality control, we assume that they have no systematic error, i.e., that their bias is zero. We also assume the sampling error to be small, so that the uncertainty $M$ associated with the measurement error can be well approximated by the precision of the employed measurement methodology (Dickson et al., 2007).

The **representation error**, $R$, is a result of the fact that we develop our statistical model (GRaCER) on a grid that is in many places coarser in time and space than the typical scales of variability of TA and $pCO_2$. As a result, any given observation may not be representative for the $1° \times 1°$ monthly grid cell used as a basis for our regression, leading to a bias in the estimated mean relative to the true spatial and temporal mean. This problem is particularly severe if the number of samples within any grid cell is low, and the spatio-temporal variability is high. This is often the case. For example, more than 90% of the data in the monthly gridded $pCO_2$ product are based on a single day of sampling within the month. The situation is more dire for TA for which there are 10-fold fewer observations than $pCO_2$. Since we are lacking full knowledge of the spatial and temporal variability of TA and $pCO_2$, we cannot fully quantify the representation error. Instead, we approximate it using the few regions where we have sufficient observations, or then using closely-related parameters for which we have more observations. For simplicity, we make the assumption that the uncertainty associated with this error is, on global average, normally distributed with a bias of 0.

The uncertainty associated with the **prediction error**, $P$, is determined by the test scores from the evaluation of the statistical model vis-à-vis the independent test data. The test scores describe the error incurred in the prediction of the subset of data that is not used in the training step. This error includes also the propagated uncertainty associated with the predictor variables.

We summarize these uncertainties with mean biases ($\frac{\sum_{i=1}^{N}(\hat{y}_i - y_i)}{N}$) and root mean squared error (RMSE, $\sqrt{\frac{\sum_{i=1}^{N}(\hat{y}_i - y_i)^2}{N}}$), where $y$ is the target value, $\hat{y}$ is the predicted value and $N$ is the number of observations. We separate the coastal and open ocean regions using the COastal Segmentation and related CATchments (COSCATs) mask (Laruelle et al., 2013) in order to reflect their very different levels of spatio-temporal variability.

### 3.1.1 Uncertainty for total alkalinity

We adopt an uncertainty $M$ associated with the measurement error of TA of $\pm 5$ $\mu$mol kg$^{-1}$ based on the laboratory intercomparison by Bockmon and Dickson (2015). This is only half the accuracy of $\pm 10$ $\mu$mol kg$^{-1}$ or 0.5% reported by GLODAPv2

**Table 3.** Summary of the uncertainties of total alkalinity and $p$CO$_2$ from the different error sources (see Table 2) separately evaluated for the open ocean and for coastal regions (defined by the COSCATs regions Laruelle et al. 2013). See text for details on how the different sources were quantified.

| | Alkalinity ($\mu$mol kg$^{-1}$) | | $p$CO$_2$ ($\mu$atm) | |
|---|---|---|---|---|
| Uncertainty | Open ocean | Coastal | Open ocean | Coastal |
| Measurement | 5 ($\leq 10$) | | 2 ($\leq 5$) | |
| Representation | 16 | 34 | 7 | 17 |
| Prediction | 13 | 28 | 12 | 27 |
| Total | 21 | 45 | 14 | 32 |

(Olsen et al., 2019). We consider this to be an overly-conservative estimate, since Bockmon and Dickson (2015) pointed out that the majority of the laboratories involved in the round-robin exercise achieved an accuracy of better $\pm 5$ $\mu$mol kg$^{-1}$. We thus opted for this lower value that is more representative of the majority of the data.

Owing to the sparseness of the TA observations, we cannot estimate the uncertainty $R$ associated with the representation error directly. Instead, we use the high correlation between TA and salinity. This permits us to determine the representation error for TA indirectly from an estimate of the representation error of sea-surface salinity. Concretely, we compare the test RMSE of TA predicted with GLODAPv2's in situ salinity with the RMSE of TA predicted with the satellite-based SODA salinity (see Table 1). Since the latter salinity is supposed to reflect the true time-space average over each grid cell, the difference between these two salinities is a direct estimate of the uncertainty associated with the representation error for salinity. Consequently, the difference in TA from these two estimates is an estimate of the uncertainty associated with the representation error for TA. The resulting estimates for the open and coastal ocean are summarized in Table 3.

The uncertainty $P$ associated with the prediction error is based on the model's RMSE score calculated from test data and is listed in Table 3 and Figure 3a. The global mean prediction error for the open ocean amounts to 13 $\mu$mol kg$^{-1}$, with some regional differences. The prediction error is more than twice this number in the coastal regions, i.e., 28 $\mu$mol kg$^{-1}$ (coastal regions are defined by the COSCATs regions Laruelle et al. 2013). We find especially high prediction errors, for example, in the highly dynamic Amazon outflow region or the Gulf of Maine in the northwestern Atlantic. However, in such regions, one can expect that part of the high prediction error is actually stemming from a representation error, as we are not using directly co-measured variables when we train our regression model. In Figure A2, we show the spatially and climatologically mapped test errors for TA in Figure A2 using the GRaCER approach.

While the global bias of the TA product of OceanSODA-ETHZ is near zero (0.5 $\mu$mol kg$^{-1}$), confirming our assumption about the unbiased nature of our prediction error, this is not the case regionally. For example, OceanSODA-ETHZ tends to consistently overestimate TA in the southeastern Atlantic and underestimate TA in the southern Indian Ocean (Figure 3b). A seasonal breakdown of the biases into DJF and JJA reveals that the winter period of each hemisphere has biases in the high latitudes, though the paucity of data means that we can place less weight on this finding.

A good check on the model prediction error is provided by comparing the estimated TA against independent observations. To this end, we use data from the Hawaii Ocean Time-series (HOT), the Bermuda Atlantic Time Series (BATS) and the Irminger station shown in Figures 4(*a,b,e,f,i,j*) and Table 4. For the period 1990–2018, the bias for BATS is 3 $\mu$mol kg$^{-1}$ and for HOT -2 $\mu$mol kg$^{-1}$, indicating that the method captures the overall structure and variability of TA well at these subtropical stations. Further, the mean seasonal cycle is relatively well represented at HOT and BATS, being within one standard deviation of the interannual variability when averaged as a climatology (Figure 4b,f). However, the results are not as good for the Irminger station in the Atlantic high latitudes ($\sim$65°N), where OceanSODA-ETHZ has a large negative bias (-10 $\mu$mol kg$^{-1}$) when compared to TA computed from the observed $p$CO$_2$ and DIC. OceanSODA-ETHZ also overestimates the weak seasonal cycle of TA at the Irminger station, contributing to the large bias that is particularly strong from December to May. The RMSE at Irminger station is 15 $\mu$mol kg$^{-1}$, less than 5 $\mu$mol kg$^{-1}$ larger than the RMSE for HOT and BATS stations (10 $\mu$mol kg$^{-1}$ respectively) owing to the small interannual and seasonal amplitude at Irminger. The RMSEs are thus smaller than the mean prediction error (13 $\mu$mol kg$^{-1}$) at the subtropical stations, yet exceeds this mean estimate at the high latitude station.

### 3.1.2   Uncertainty for $p$CO$_2$

For the uncertainty $M$ associated with the measurement error of $p$CO$_2$, we adopt a value of $\pm 2$ $\mu$atm. This reflects the fact that 80% of the data we have used from SOCAT (flags A and B) have a precision better than that number and an accuracy of similar magnitude. The remaining data we used (SOCAT flags C and D) have a measurement precision and accuracy of less than $\pm 5$ $\mu$atm.

We estimate the uncertainty $R$ associated with the representation error of $p$CO$_2$ on the basis of a spatio-temporal gradient analysis. To this end, we compare the $p$CO$_2$ in our regular grid that has a resolution of $1° \times 1°$ by 1 month, with the $p$CO$_2$ binned to a grid with twice this resolution, i.e., $0.5° \times 0.5°$ by 15-days. In regions with high spatio-temporal coverage, the difference in the average of adjacent grid cells represents the potential change that can occur within the coarser $1° \times 1°$ by 1 month grid cell. The spatial and temporal gradients are calculated separately and we take the average of these two elements. Using this analysis, we estimate a representation error of our $p$CO$_2$ estimates of 7 $\mu$atm and 17 $\mu$atm for the open and coastal ocean respectively (Table 3).

From the RMSE of our test data, we estimate an uncertainty $P$ associated with the prediction error of $p$CO$_2$ of 12 $\mu$atm for the open ocean and 28 $\mu$atm for the coastal ocean (Table 3). Within the open ocean (Figure 3e), the eastern tropical Pacific has the highest RMSE, but this is also the region with the highest variance in the observations. The strong horizontal gradients in the region increase the errors, particularly at cluster boundaries. The high RMSE for the coastal region stems primarily from coastal Antarctica as well as some coastal regions in the higher latitudes of the northern hemisphere. The former is due to large uncertainties in $p$CO$_2$ during the summer months when retreating ice and ensuing rapid net primary production result in large gradients (Bakker et al., 2008). The climatologically mapped errors for $p$CO$_2$ are shown in Figure A2(b,d) created by mapping the test errors to the clusters and averaged over the ensemble.

The comparison between the regression estimated and observed $p$CO$_2$ reveal regional biases (Figure 3e), despite the global bias being close to zero (-0.37 $\mu$atm). Some of the highest biases are found, again, in the eastern tropical Pacific, where strong

**Table 4.** Comparison of training and independent data sources with various methods for the open ocean region using the COSCATs coastal mask by Laruelle et al. (2013). GLODAP refers to the GLODAP v2 2019 data (Olsen et al., 2019), HOT to the Hawaii Ocean Time-series (Dore et al., 2009), BATS to Bermuda Atlantic Time Series (Bates and Peters, 2007), SOCAT is the 2019 version of the Surface Ocean Carbon Atlas (Bakker et al., 2016), LDEO is the Lamont-Doherty Earth Observatory $pCO_2$ data set for points not present in SOCAT (Takahashi et al., 2019), SOCCOM is the pH measured by autonomous floats from the Southern Ocean Carbon and Climate Observations and Modeling project (Johnson et al., 2016). Statistical outliers were excluded in the calculation of LDEO RMSE. OS-ETHZ is the OceanSODA-ETHZ data from this study, NNGv2 is from Broullón et al. (2018), LIARv2 from Carter et al. (2018), CMEMS-FFNNv2 from Denvil-Sommer et al. (2019), and SOMFFN from Landschützer et al. (2016). NNGv2 and LIARv2 predictions are made with SODA salinity and OSTIA sea surface temperature resulting in different estimates to the original publications (Broullón et al., 2018; Carter et al., 2018). Note that the full data set is used for OceanSODA-ETHZ, unlike Table 3 which presents the errors for test years.

| | | TA ($\mu$mol kg$^{-1}$) | | | $pCO_2$ ($\mu$atm) | | DIC ($\mu$mol kg$^{-1}$) | | | pH | |
|---|---|---|---|---|---|---|---|---|---|---|---|
| | | GLODAP | HOT | BATS | SOCAT | LDEO | GLODAP | HOT | BATS | GLODAP | SOCCOM |
| Bias | **this study** | 0.5 | -2.1 | 2.6 | -0.4 | 0.1 | 0.5 | -1.0 | 0.4 | -0.001 | 0.009 |
| | LIAR + FFNN | 0.3 | -3.1 | 0.2 | 0.5 | 0.6 | | | | 0.001 | 0.013 |
| | NNGv2 | 1.2 | -3.2 | 4.3 | | | 2.3 | 2.2 | -0.4 | | |
| | SOMFFN | | | | 0.4 | 0.4 | | | | | |
| RMSE | **this study** | 17.5 | 9.5 | 10.1 | 11.1 | 19.9 | 16.3 | 8.7 | 9.1 | 0.024 | 0.036 |
| | LIAR + FFNN | 18.0 | 8.8 | 8.8 | 13.1 | 19.6 | | | | 0.023 | 0.037 |
| | NNGv2 | 16.2 | 6.7 | 10.4 | | | 23.1 | 9.5 | 15.2 | | |
| | SOMFFN | | | | 11.7 | 21.4 | | | | | |
| $r^2$ | **this study** | 0.91 | 0.58 | 0.13 | 0.82 | 0.45 | 0.93 | 0.77 | 0.76 | 0.67 | 0.047 |
| | LIAR + FFNN | 0.91 | 0.6 | 0.21 | 0.78 | 0.44 | | | | 0.67 | -0.043 |
| | NNGv2 | 0.93 | 0.75 | -0.1 | | | 0.82 | 0.71 | 0.38 | | |
| | SOMFFN | | | | 0.82 | 0.49 | | | | | |

horizontal gradients drive the observed juxtaposed biases. The large negative biases in winter in the Southern Ocean are likely driven by the paucity of data in this region (Gregor et al., 2019; Gray et al., 2018; Bushinsky et al., 2019).

The time series comparisons show that the seasonal cycle is well represented at BATS and HOT with $r^2$ scores of 0.89 and 450 0.82 respectively (Figures 4d,h). Low biases (< 2 $\mu$atm in absolute terms) further demonstrate that $pCO_2$ estimates are reliable in the subtropics. The seasonal cycle is also well captured at the Irminger station in the high latitudes, but a lower $r^2$ score and larger bias (-8.0 $\mu$mol kg$^{-1}$) allude to the dampened amplitude of the seasonal cycle particularly in the winter months (Figure 4).

### 3.2 Uncertainties of the calculated parameters

We determine the uncertainties of the calculated parameters in two ways. First, we propagate the uncertainties of $pCO_2$ and TA through pyCO2SYS (Orr et al., 2018; Humphreys et al., 2020) onto the computed parameters DIC, pH, and $\Omega$. This yields an

expected uncertainty that we refer to as a "bottom-up" total uncertainty estimate. Second, we obtain a "top down" uncertainty estimate, by comparing the calculated $DIC$ and pH with independent measurements (see Table 4). We first describe the top-down estimates and then the bottom-up. For the top-down estimate, we use the GLODAP $DIC$ and pH data as our independent

test of the method's performance. This is because these data are not used in any way in our estimation.

In the global mean, the computed $DIC$ in OCEANSODA-ETHZ has a very low bias compared with in situ GLODAP measurements (0.5 $\mu$mol kg$^{-1}$) (Table 4). Spatially, the $DIC$ biases reveal a more nuanced picture, with large positive biases in the western equatorial Pacific and negative biases in the western equatorial Atlantic. The bias in the western equatorial Atlantic matches the negative bias in the $p$CO$_2$ in the same region; however, the source of the $DIC$ bias in the Pacific is not

clear from the $p$CO$_2$ and TA test data biases. The global mean of the top-down uncertainty estimate for $DIC$ (16.3 $\mu$mol $^{-1}$ for the open ocean) is then perhaps a better reflection of the uncertainty as positive and negative values do not cancel each other out.

It is interesting to point out that the computed $DIC$ in OceanSODA-ETHZ compares very favorably to directly estimated $DIC$ products, such as that provided by NNGv2. Our uncertainty associated with the prediction error for $DIC$ of 16.3 $\mu$mol

$^{-1}$ is substantially better than theirs (23.1 $\mu$mol kg$^{-1}$) across all independent data.

The comparison of the $DIC$ time series data (BATS, HOT and Irminger stations) supports the findings of the global top-down estimates (Figure 6). The biases are relatively low for HOT and BATS (-1.0 and 0.4 $\mu$mol kg$^{-1}$ respectively), but the bias is much larger at Irminger station (-6.9 $\mu$mol kg$^{-1}$) which is at a much higher latitude (64°N) compared to HOT and BATS in the subtropics (< 35°N). However, this bias is not reflected in the zonal average of the seasonal biases (Figure 5c). Similarly,

the RMSE is also larger at Irminger station (14 $\mu$mol kg$^{-1}$) compared to HOT and BATS ($\sim$9 $\mu$mol kg$^{-1}$) in the subtropics). Despite these differences in the top-down error, the same amount of variability is represented by the OCEAN-SODA $DIC$ for all three stations ($\sim$0.76) owing to the larger seasonal cycle at Irminger station.

The pH comparison with the GLODAP pH measurements shows that OceanSODA-ETHZ has a negligible bias (0.001). As with $DIC$, regional biases in pH re larger than the global average, with the coastal and high latitude oceans contributing

significantly to the regional biases. The RMSE of pH with respect to GLODAP is also low (0.024), but is slightly outperformed by the RMSE of pH calculated with LIARv2 TA and FFNNv2 $p$CO$_2$ (made available in the FFNNv2 data set, 0.023). But given the uncertainty of GLODAP pH (0.01), the difference is negligible.

For the "bottom-up" estimate, we propagate the total uncertainty of $p$CO$_2$ and only the uncertainties associated with the measurement and prediction errors for TA. We do this to avoid including the representation error twice in the bottom-up

estimate, as we hypothesize that the representation error of TA is largely accounted for by the representation error of $p$CO$_2$ and vice versa. We choose to use the representation error of $p$CO$_2$ rather than TA as the larger number of samples gives us greater confidence in the estimate. Moreover, we feel that the assumptions we make in the estimate of the TA representation error are larger than those for $p$CO$_2$, thus further justifying our choice in using only the representation error of $p$CO$_2$.

The comparison of top-down vs bottom-up uncertainty estimates for open and coastal oceans is shown in Figure 7 with values

of the total uncertainties also shown. The top-down and bottom-up uncertainty estimates for $DIC$ are relatively accurate with the estimates being within 5% of each other in the open ocean and 9% in the coastal ocean. However, the estimates are not as

coherent for pH where the bottom-up error is 23% smaller than the top-down uncertainty in the open ocean. The difference is even bigger in the coastal ocean where the bottom-up is 31% smaller than the top-down uncertainty.

## 4  Results

### 4.1  Comparison with other climatologies

A spatial comparison between OceanSODA-ETHZ and existing products might reveal potential biases in our product if the bias is present in all comparisons. We compare TA against LIARv2 and NNGv2 (Carter et al., 2018; Broullón et al., 2018) by first taking the difference for each month and then calculating the average for these differences. The same approach is used to compare $pCO_2$ with SOMFFN and FFNNv2 (Landschützer et al., 2016; Denvil-Sommer et al., 2019).

The differences in TA between OceanSODA-ETHZ and NNGv2 and LIARv2 are on the same order of magnitude in the open ocean as the prediction error (13 $\mu$mol kg$^{-1}$) but are slightly larger on average for NNGv2 than for LIARv2 (Figure 8a,b). There is also some agreement in the spatial pattern of the distribution, particularly in data sparse parts of the Pacific and Indian Oceans. The larger differences may stem from the fact that both LIARv2 and NNGv2 are more constrained by spatial coordinates than GRaCER. NNGv2 uses latitude and longitude as predictors while the LIARv2 approach interpolates the linear regression coefficients for every 5° grid cell (Broullón et al., 2018; Carter et al., 2018). The divergence in data sparse regions is thus not surprising. Though, it must be emphasized that this comparison serves more as a "sanity check" than as a ground truthing exercise.

The differences between OceanSODA-ETHZ $pCO_2$ and FFNNv2 and SOMFFN are smaller and not as spatially coherent than those for TA. The differences are marginally larger for the OceanSODA - FFNNv2 than for OceanSODA - SOMFFN (Figure 8c,d). This is consistent with the data in Table 4, where the metrics for the SOMFFN are very similar to OceanSODA-ETHZ. The smaller difference between the latter should not come as a surprise as the GRaCER approach is built on the two-step cluster-regression approach of the SOMFFN, while the FFNNv2 approach includes spatial coordinates (Landschützer et al., 2016; Denvil-Sommer et al., 2019). In general, the dissimilarity between the differences is encouraging as it indicates that OceanSODA-ETHZ is not consistently biased relative to SOMFFN and FFNNv2.

We also show the temporal evolution of the basin-mean differences between OceanSODA-ETHZ $pCO_2$ and other gap-filling methods (Figure 9). In the Atlantic (Figure 9a), OceanSODA-ETHZ $pCO_2$ is $< 2$ $\mu$atm lower than the mean of the other gap-filling methods for the period 1990 to 2008. Thereafter, the difference is $< 1$ $\mu$atm. In the Indian ocean, our $pCO_2$ estimates have a persistent negative difference of $\sim 2$ $\mu$atm (Figure 9c). The comparison in the Pacific (Figure 9b) is the most consistent with the other methods, with a slight positive difference in the beginning of the period (pre-1990). The OceanSODA-ETHZ estimates of $pCO_2$ in the Southern Ocean (Figure 9d) have a large positive difference prior to 1990 – up to 6 $\mu$atm for one of the ensemble members. This difference quickly diminishes and is near zero by 1990. There is also a negative difference later in the period (2004 to 2015); however, the ensemble spread over this period is large.

The comparison with other methods illustrates that while gap-filling methods are converging on a global scale, there are regionally differences. Further, large differences in $p$CO$_2$ between methods prior to 1990 indicates high uncertainty for this period.

## 4.2 Seasonal Climatologies

The climatological mean spatial distribution of TA, $p$CO$_2$, DIC, pH, and $\Omega$, obtained by averaging the estimates from 1985 through 2018, reveal a very rich and diverse pattern of variability with commonalities and differences (Figure 10). The climatological maps are accompanied by Hovmoeller diagrams that show the zonal average of the seasonal cycle for each of the variables (Figure 11). We also show climatological time series for each of the variables at high (55°N, 170°W), mid (30°N, 170°W) and low (10°N, 170°W) latitude locations.

Total alkalinity shows the largest differences between basins, with the mean alkalinity being much higher in the saltier Atlantic than in the Pacific and Indian basins (Boutin et al., 2018) (Figure 10a). The spatial variability of TA on a global scale exceeds the variability of the seasonal cycle (Figure 11a,e). Seasonal variability of TA is on the order of 20 $\mu$mol kg$^{-1}$ at the chosen mid and high latitude locations, while the latitudinal gradient is as large as 150 $\mu$mol kg$^{-1}$. However, much of the TA seasonality is driven by seasonal changes in salinity due to precipitation and ice melt in the respective regions.

Dissolved inorganic carbon is more homogeneous across the basins, but has a much larger meridional gradient than TA, amounting to more than 150 $\mu$mol kg$^{-1}$ (Figure 10b). This meridional gradient is seasonally substantially more modified than is the case for TA, particularly in the high latitudes of the northern hemisphere where the seasonal cycle is as large as 100 $\mu$mol kg$^{-1}$ 11b,g). The larger seasonal cycle in DIC is due to the carbon uptake by spring-time phytoplankton blooms and stratification during the warmer seasons (Siegel et al., 2002). The magnitude of the spring-time blooms is dampened by iron limitation in the Southern Ocean, visible by a smaller seasonal cycle amplitude ($\sim$40 $\mu$mol kg$^{-1}$) (Watson et al., 2000; Tagliabue et al., 2017). However, the background DIC concentration in the Southern Ocean is much larger due to upwelling of DIC-rich circumpolar deep waters driven by the persistent westerlies south of 50°S (Marshall and Speer, 2012).

The spatial distribution and seasonal cycles of $p$CO$_2$ and pH (Figures 10c,d and 11c,d,h,i) are strongly negatively correlated due to the inverse stoichiometric relationship between dissolved aqueous CO$_2$ and [H$^+$] (Dickson et al., 2007). The reduction in DIC is concomitant with the reduction in $p$CO$_2$ in the high latitudes due to the biological uptake of [CO$_2$] (Figure 11g,h). However, this relationship does not hold true in the mid latitudes, where the slight decrease in DIC is contrasted by a relatively strong increase in $p$CO$_2$. This is due to the positive temperature dependence of $p$CO$_2$ (the opposite is true for pH) (Takahashi et al., 1993), which will be elaborated on in the discussion.

The spatial distribution of $\Omega$ (Figures 10e,f and 11e,j) strongly reflects the concentration of the carbonate ion, which can be well approximated by the difference between TA and $DIC$ (Sarmiento and Gruber, 2006). Given that the seasonal cycle of TA is much weaker than $DIC$, the latter dominates the seasonal cycle of $\Omega_{\text{Ar}}$ (Figures 11e,j). This would also be true for $\Omega_{\text{Ca}}$ which only differs from $\Omega_{\text{Ar}}$ in magnitude and not in distribution or seasonality (the latter is not shown).

## 4.3 Global and basin-scale long-term trends

The OceanSODA-ETHZ data set can provide important novel constraints on the long-term trends in ocean acidification. We determine the long-term trends by a linear regression approach, restricting the period to 1990 through 2018, thus leaving out the 1980s, where the estimates are much more uncertain.

The global and basin-scale trends for $pCO_2$ are remarkably similar (varying by only about 0.5 $\mu$atm decade$^{-1}$ around the global mean of 16.5 $\mu$atm decade$^{-1}$) (Table 5. The ocean trends are also very consistent with the trends in atmospheric $CO_2$ ($\sim$ 18.6 $\mu$atm decade$^{-1}$), reflecting the fact that the $CO_2$ system in the surface ocean is following the increase in the atmosphere very closely globally as well as in all ocean basins. The atmospheric trend is slightly steeper than those in the ocean. This is as expected since the atmospheric $CO_2$ increase forces the ocean, leading to a slight delay, a reflection of an increase in the air-sea disequilibrium over time (see e.g., discussion in Matsumoto and Gruber (2005)). This growth in the air-sea disequilibrium is the driving force behind the increase in the oceanic sink strength for anthropogenic $CO_2$ over time (Gruber et al., 1996).

The basin-scale consistency holds true for pH as well (-0.016 units decade$^{-1}$) and $\Omega_{ar}$ (-0.07 units decade$^{-1}$), where global values are very similar to those averaged across the different ocean basins. In contrast, there is more spatial variability in the $DIC$ trends, but the overall magnitude of about 9 $\mu$mol kg$^{-1}$dec$^{-1}$ largely represents the uptake of anthropogenic $CO_2$ from the atmosphere. The spatial differences in the $DIC$ trends are also mirrored in the spatial variations in the TA trends, with regions with higher $DIC$ trends having higher TA trends. This makes sense in terms of positive trends in TA increasing the buffering capacity, hence permitting DIC to grow faster for the same increase in seawater $pCO_2$. The trends in TA themselves are driven almost entirely by salinity with a basin-scale correlation of 0.99 (see Cheng et al. (2020)).

**Table 5.** Linear trends and their standard errors for OceanSODA-ETHZ variables for the period 1990 to 2018. All columns show increases per decade (dec$^{-1}$). All trends in the table are significant ($P < 0.05$). We exclude the Arctic as the OceanSODA-ETHZ product only covers 23% of this region and may thus give spurious trends. The Ocean basins are defined by the map shown in Figure A4.

|  | TA | DIC | $\Omega_{ar}$ | pH | $pCO_2$ | $pCO_2^{atm}$ |
|---|---|---|---|---|---|---|
|  | $\mu$mol kg$^{-1}$dec$^{-1}$ | $\mu$mol kg$^{-1}$dec$^{-1}$ | units dec$^{-1}$ | units dec$^{-1}$ | $\mu$atm dec$^{-1}$ | $\mu$atm dec$^{-1}$ |
| Global | 1.5 $\pm$ 0.1 | 8.6 $\pm$ 0.1 | -0.07 $\pm$ 0.00 | -0.016 $\pm$ 0.000 | 16.5 $\pm$ 0.1 | 18.6 $\pm$ 0.1 |
| Atlantic | 3.1 $\pm$ 0.2 | 10.0 $\pm$ 0.4 | -0.07 $\pm$ 0.00 | -0.016 $\pm$ 0.000 | 16.7 $\pm$ 0.2 | 18.8 $\pm$ 0.2 |
| Pacific | 0.5 $\pm$ 0.1 | 8.1 $\pm$ 0.3 | -0.07 $\pm$ 0.00 | -0.016 $\pm$ 0.000 | 16.7 $\pm$ 0.1 | 18.7 $\pm$ 0.1 |
| Indian | 4.4 $\pm$ 0.4 | 10.9 $\pm$ 0.6 | -0.06 $\pm$ 0.00 | -0.015 $\pm$ 0.000 | 16.2 $\pm$ 0.4 | 18.3 $\pm$ 0.1 |
| Southern | 0.5 $\pm$ 0.1 | 7.0 $\pm$ 0.6 | -0.06 $\pm$ 0.01 | -0.017 $\pm$ 0.001 | 16.0 $\pm$ 0.4 | 18.5 $\pm$ 0.1 |

## 5 Discussion

### 5.1 Choosing the appropriate machine learning configuration

Here we consider two notable decisions that have a large impact on the final estimates: 1) the use of the ensemble approach, and 2) the choice of regression algorithm. For details on the minor choices, see section A3 in the supplementary materials.

As previously motivated, we opt for the cluster-regression approach that is able to generalize estimates in sparse regions due to information sharing within a cluster. However, cluster boundaries are often semi-discrete, resulting in artifactual boundaries. This makes the output of cluster-regression approaches less suitable for studies where gradients over short time periods or

580 distances are assessed, e.g., for the detection of extreme events. Our approach removes these boundaries and improves the robustness of the estimates by eliminating, to large extent, the sensitivity of the regression to the clustering algorithm.

The second major consideration is the choice of regression algorithm. Our choice of different algorithms for TA and $pCO_2$ may seem peculiar. However, this decision was informed by the nature of the problems. While regressing TA and $pCO_2$ are conceptually similar, the size of each data set and the distribution of training samples sets them apart. When gridded to a

585 monthly by $1° \times 1°$ resolution, the number of data are $\sim 300\,000$ for SOCAT $pCO_2$ and $\sim 16\,000$ for GLODAP TA; *i.e.*, nearly 20 times more data for the former. However, the strong linear correlation between TA and salinity (r $= 0.96$) compensates for the poor sampling distribution; that is, as long as the regression method is able to extrapolate — a criterion that the support vector regression (SVR) method meets.

$pCO_2$ is poorly correlated to any proxy, suggesting that the regression problem is more complex, thus requiring a method

that is appropriately non-linear. Gradient-boosted decision trees (GBDT) and feed-forward neural-networks (FFNN) meet this criterion. But why not just use one of these approaches? Work by Gregor et al. (2019) found that an ensemble of methods (SVR, GBDT and FFNN) outperformed each individual member. And while SVR performed well in Gregor et al. (2019), the method does not scale to larger problems, which GBDT and FFNNv2 are capable of, leading to our choice of the latter two. One critique of GBDT in this application may be that, being a tree-based method, it is not able to extrapolate. However, we feel that

the cluster-regression approach combined with the large number of training data for $pCO_2$ compensates for this shortcoming. GBDT provide also provide useful diagnostics, such as feature importances, that, when combined with the GRaCER approach, provides useful information about the spatial and seasonal importance of the proxies.

### 5.2 Why are pH uncertainties less well-constrained?

One of the novel contributions of this study is that we are able to assert the validity of our results by comparing the bottom-up

(propagated) with the top-down uncertainties (in situ comparisons). Using this approach, we show that the uncertainty estimates of $DIC$ are remarkably well-constrained, with the top-down being within 5% of the bottom-up uncertainty estimate for the open ocean (7b). The same statistics for pH yields a 23% difference between the two budgets. The question is thus, why are pH bottom-up and top-down uncertainty estimates much less consistent?

To assess this problem from the bottom-up perspective, we need to consider the uncertainties of $p\text{CO}_2$ and TA. But given that the $DIC$ uncertainty budgets are well constrained, we can, with some certainty, rule out the bottom-up uncertainty estimate as the source for the larger mismatch in pH.

The source of the mismatch must thus be driven primarily by uncertainties in the top-down perspective, where it may be that the representation error of pH is larger than for $DIC$. We rule out the measurement error as a contributor to the mismatch, as the bias of the measurements (provided accurate calibration to reference samples) should be normally distributed around zero. Thus, the representation error is the most likely candidate, due to the temperature and pressure sensitive nature of pH compared to the conservative nature of $DIC$ with respect to the same variables (Dickson et al., 2007). This is important given that our "surface" pH data from GLODAP can be as deep as 30 m, while $p\text{CO}_2$ is typically measured at the intake depth of most ships, which is typically at 5 m (Bakker et al., 2016). A basic sensitivity study of the variability of pH in the surface layer shows that the median standard deviation of "surface" pH at a single station is 0.004 units (even when normalized to a standard temperature and depth). The same approach for $DIC$ yields a standard deviation of 2 $\mu\text{mol kg}^{-1}$. This suggests that the vertical mismatch error of GLODAP alone already explains a good deal of the discrepancy in the uncertainty budgeting. One might reduce this uncertainty by placing tighter constraints on the definition of "surface" pH, limiting the depths between 3 m and 15 m, for example, but at the loss of valuable test samples. The representation error (described in Section 3.1) could further explain the remaining disparity in the pH error budget.

## 5.3  Can we reduce the total uncertainty?

The last two decades have seen major improvements in the accuracy and precision of the TA and $p\text{CO}_2$ measurements, leading to substantial reductions in the measurement errors. The introduction of certified reference materials for TA and a standardized approach for measuring $p\text{CO}_2$ with reference gases means that the uncertainties associated with the measurements are low.

In contrast, the prediction uncertainty is the largest contributor to the total uncertainty for both $DIC$ and TA, suggesting that this could be a fruitful avenue to pursue. However, current literature suggests that this is unlikely. Gregor et al. (2019) showed that within a selection of six gap filling methods, all achieved similar accuracy scores when compared with independent data, upon which the authors suggested that we have hit a "wall".

This leaves the representation error, which contributes a moderate fraction to the total $p\text{CO}_2$ and TA uncertainties in the open ocean and even less in the coastal ocean. A back-of-the-envelope calculation shows that increasing the resolution of $p\text{CO}_2$ fourfold (from monthly by 1°x1° to 8-daily by 0.25°x 0.25°) could decrease the representation error by 2.5 $\mu$atm ( 35%) for the open ocean and 3.2 $\mu$atm for the coastal ocean ( 20%). This is perhaps not as much as expected, but these small gains are larger than those that are currently being made with regard to the prediction errors (Gregor et al., 2019). This is not applicable for TA, where ungridded values are already used to train GRaCER and where the selection of the predictor variables are likely more important.

Why are these gains smaller than hoped? It may be that our gradient approach for calculating the representation error breaks down as the resolution increases due to the decreasing number of adjacent grid points. This is hardly surprising considering that 78% of grid cells in the SOCAT v2019 monthly-gridded product are represented by sampling on a single day that falls within

that period (Bakker et al., 2016). Another possibility is that decreasing both the spatial and temporal resolution exposes the sharp mesoscale gradients that are otherwise averaged over at larger resolutions (Resplandy et al., 2009; Monteiro et al., 2015). With the currently available data, it seems as if the reduction to be made in the total uncertainty by increasing the resolution of the prediction data will be small.

## 5.4 Regional sensitivity of $p\text{CO}_2$ to driver variables

Here we demonstrate one of the possible ways in which the OceanSODA-ETHZ data can be used to gain further insight into the marine carbonate system.

We decompose and attribute the mean seasonal cycle variability of $p\text{CO}_2$ to its drivers, namely TA, $DIC$, temperature and salinity. Past studies using observation based-products have been limited to a simpler thermal/non-thermal decomposition of $p\text{CO}_2$ due to the lack of $DIC$ and TA (Takahashi et al., 2002; Landschützer et al., 2015). This is thus the first time that a full decomposition of $p\text{CO}_2$ has been applied to observation-based data for a global domain. The decomposition is performed with *pyCO2SYS* by keeping all but one of the drivers constant (to the average) and assess the influence on $p\text{CO}_2$ (Humphreys et al., 2020). This is similar to previous studies that applied an analogous decomposition to ocean simulation output (Lovenduski et al., 2007; Fassbender et al., 2018; Gallego et al., 2018). An important note to make, considering that we intend that OceanSODA-ETHZ is used for ocean acidification studies, is that the decomposition of pH would result in virtually the same contribution of the drivers (Dickson et al., 2007).

The seasonal amplitude of $p\text{CO}_2$ is driven predominantly by changes in $DIC$ in the high and equatorial latitudes, and by temperature in the mid latitudes (Figure 12d). At first glance, these results may seem similar to those that the simpler thermal decomposition might result in, requiring only temperature. However, in regions where the seasonal amplitude of $p\text{CO}_2$ is smaller, the importance of TA becomes more apparent. For example, at the mid-latitude station (Figure 12b), TA and $DIC$ synergistically act to dampen the impact of temperature on $p\text{CO}_2$. Conversely, at the equatorial station (Figure 12c) the effect of TA on $p\text{CO}_2$ opposes that of the more dominant $DIC$. Further, there are regions in the tropics where TA is the dominant driver due to the weak seasonal cycle of both temperature and $DIC$.

## 5.5 Recommendations for use

In order to use the OceanSODA-ETHZ product in an optimal manner, it is important to be aware of its strengths and weaknesses.

The primary use of the OceanSODA-ETHZ data set is to determine and assess the seasonality, the interannual variations and trends of ocean acidification thanks to its containing all relevant parameters of the marine carbonate system (Landschützer et al., 2015, 2016; Gregor et al., 2018; Keppler and Landschützer, 2019). However, users of the OceanSODA-ETHZ product should be aware of the fact that that data prior to the 1990's should be treated with care due to the paucity of SOCAT $p\text{CO}_2$ training data during this period (Rödenbeck et al., 2015; Watson et al., 2020). This was recently demonstrated by Watson et al. (2020) who used an ensemble of various regression approaches to show that the spread of $p\text{CO}_2$ estimates prior to the 1990's

is large due to the paucity of data. Similarly, Gregor et al. (2019) showed that $p\text{CO}_2$ estimates prior to 1990 tend to have a slightly positive bias.

The product is also very well suited for assessing models. Thanks to the spatially resolved estimates of uncertainty for TA and $p\text{CO}_2$ (Figure A2), one cannot only assess the model-observation mismatches, but also weigh them with the appropriate uncertainties.

A strength of the OceanSODA-ETHZ product is that it extends further into the coastal margin than most previous studies (Iida et al., 2015; Landschützer et al., 2016; Denvil-Sommer et al., 2019). This is achieved i) by including coastal observations during the training, and ii) by using a larger number of clusters compared to other clustering approaches (Landschützer et al., 2016; Watson et al., 2020). This permits to better separate open ocean and coastal variability through the inclusion of suitable variables in the clustering step (e.g. Chl-a for $p\text{CO}_2$, and see Figure A5 to see a representation of cluster boundaries). This

gives us confidence in the coastal estimates of the climatological seasonal cycle. Our product is therefore comparable to that of Landschützer et al. (2020) who blended separate coastal and open ocean $p\text{CO}_2$ products into a single climatological product with monthly resolution (Landschützer et al., 2016; Laruelle et al., 2017).

The total uncertainties of our estimates in the coastal ocean are considerably larger compared to the open ocean estimates (Figure 7). This reflects the much higher spatio-temporal variability of the physical and chemical environment in the coastal

ocean, leading to much higher variations in the marine carbonate system (Laruelle et al., 2017). Since our predictor variables are only partially reflecting this variability, a large portion of the high total uncertainty is due to a high representation error (Table 3). Increasing the resolution of the products may improve the total uncertainty of coastal estimates as done by Laruelle et al. (2017). Until we arrive at this point, the OceanSODA-ETHZ data should be used with care in the coastal ocean. Further, we recommend that researchers interested in the investigation of interannual variability and trends in the coastal ocean using

the OceanSODA-ETHZ product should also look a the underlying in situ data to gain a better understanding of the variability, trends, and uncertainties for the coastal region of interest.

## 6 Summary

Our approach for estimating TA and $p\text{CO}_2$ is an evolution of the cluster-regression approach: We create an ensemble of estimates by repeating the cluster-regression step multiple times, each with a different variation of clustering. We call this

approach the Geo-spatial Random Cluster Ensemble Regression (GRaCER). The result is an estimate that is more robust with better generalization and the output does not have the discrete cluster boundaries that single member cluster-regression approaches have.

We find that our estimates of TA are within the ballpark of previous methods with a prediction error (root mean square error) of 13 $\mu\text{mol kg}^{-1}$ for open ocean estimates, while biases are $< 1$ $\mu\text{mol kg}^{-1}$. Taking into consideration all sources of error

(measurement and representation errors), the total uncertainty is 17 $\mu\text{mol kg}^{-1}$ for TA. The prediction error for $p\text{CO}_2$ in the open ocean is 12 $\mu\text{atm}$, also with a bias of $< 1$ $\mu\text{atm}$. Including the measurement and representation errors for $p\text{CO}_2$ results in a total uncertainty of 14 $\mu\text{atm}$ for the open ocean. We estimate the total uncertainty of $DIC$ and pH to be 19 $\mu\text{mol kg}^{-1}$ and

0.022 units when compared with independent GLODAPv2 data for the open ocean. Finally, we compare the aforementioned "top-down" uncertainty estimates of $DIC$ and pH with the "bottom-up" uncertainty estimates that are calculated by propagating the TA and $pCO_2$ total uncertainy estimates through the marine carbonate system. This budgeting approach shows that we have a good grasp on the uncertainties of $DIC$ for both the open and coastal oceans. However, pH uncertainties are not as well resolved, most likely due to a mismatch in the representivity of the measured pH.

We demonstrate a use case of the OceanSODA-ETHZ data set in which we decompose the seasonal variability of $pCO_2$ into four driver components of $DIC$, TA, temperature and salinity. We find that $DIC$ is the dominant driver in the high and equatorial latitudes, while temperature contributes the majority of the signal in the subtropics. Importantly, $DIC$ and temperature are antagonistic drivers of $pCO_2$, while alkalinity always acts in opposition to the stronger of the two primary drivers. We also show the strong constraints OceanSODA-ETHZ can pose on the long-term trends in ocean acidification.

Finally, OceanSODA-ETHZ will be maintained and updated for future work.

## 7 Code and data availability

Software for the GRaCER framework is available on GitHub (access provided on request). The OceanSODA-ETHZ dataset is available at https://doi.org/10.25921/m5wx-ja34 (Gregor and Gruber, 2020).

## Appendix A: Supplement to the Methods

### A1 $pCO_2$ outlier removal

The first outlier removal method requires the $pCO_2$ to be adjusted from the ship intake temperature to the satellite SST as described by (Goddijn-Murphy et al., 2015):

$$pCO_2^{SST} = pCO_2^{SOCAT} \times \exp(0.0433 \cdot (T^{SST} - T^{SOCAT})) \tag{A1}$$

where the ship intake depth varies due to inconsistent depth between vessels and the water column state (e.g. well stratified or mixed). Here $T^{SST}$ is the foundation temperature given by the Operational Sea Surface Temperature and Sea Ice Analysis (OSTIA) product (Good et al., 2020). The OSTIA product is matched to the ungridded $pCO_2^{SOCAT}$ at daily by $0.25° \times 0.25°$ resolution (Good et al., 2020). The corrected $pCO_2$ is then binned to monthly by $1° \times 1°$ without weighting. Data are excluded where the absolute difference between $pCO_2^{SST}$ and $pCO_2^{SOCAT}$ is larger than 40 $\mu$atm.

Secondly, we exclude data that lie outside the expected ranges for the monthly climatology of $pCO_2$. The expected ranges are defined using the interquartile range outlier detection method for each pixel in a given month with the following equation:

$$IQR = Q_3 - Q_1 \tag{A2}$$
$$\text{lower limit} = Q_1 - IQR \cdot 1.5 \tag{A3}$$
$$\text{upper limit} = Q_3 + IQR \cdot 1.5 \tag{A4}$$

where $Q_1$ and $Q_3$ are the $25^{th}$ and $75^{th}$ percentiles, respectively. This approach is only applied where there are enough data present for a particular month of the year.

## A2    Target variable: $\Delta p\mathrm{CO_2}$ vs $p\mathrm{CO_2}$

In this study, one of the avenues that explored was to predict $\Delta p\mathrm{CO_2}$ instead of $p\mathrm{CO_2}$. Motivation for predicting $\Delta p\mathrm{CO_2}$ is that it might allow new measurements from recent years to add new information about the seasonal cycle in regions where sampling was previously seasonally biased, *e.g.* the SOCCOM float data (Gray et al., 2018). Bushinsky et al. (2019) showed that including the new information about the seasonal cycle of $p\mathrm{CO_2}$ in machine learning estimates resulted in stronger winter outgassing, but their results could only show this for the period that SOCCOM float data are present. One of the reasons that machine learning approaches are not able to propagate this information back through time is that the larger $p\mathrm{CO_2}$ is "anchored" by the $CO_2$ CO2 concentrations that are used as a proxy. Atmospheric $p\mathrm{CO_2}$ is required as a predictor variable to capture the interannual signal of $p\mathrm{CO_2}$. Predicting $\Delta p\mathrm{CO_2}$ might thus allow one to remove atmospheric $p\mathrm{CO_2}$ as a driver because the interannual term trend of $p\mathrm{CO_2}$ is removed.

The results appeared promising, but on further investigation we found that the regressions that were trying to predict $\Delta p\mathrm{CO_2}$ were not able to represent the increasing strength of the sink. Furthermore, we found that the interannual variability of $p\mathrm{CO_2}$ was reduced compared to results that include atmospheric $x\mathrm{CO_2}$ as a driver. Ultimately, we abandoned the approach.

## A3    Hyper-parameter selection for regression methods

### A3.1    Total alkalinity: support vector regression

Hyper-parameters for the support vector regression (SVR) were chosen on a per-cluster basis using grid search cross validation, where unshuffled K-fold cross validation with five splits was used. The $\nu$SVR variety of the algorithm from the *scikit-learn* package in Python was used. The parameters $C$, $\gamma$, and $\nu$ were selected.

### A3.2    $p\mathrm{CO_2}$: Gradient boosted decision trees

We used the *LightGBM* package to perform the gradient boosted regression with decision trees (GBDT). The GBDT algorithm was trained using early stopping, which determines the number of trees used in the model – typically one of the most important hyper-parameters. Every fifth year from 1987 to 2019 was set aside as the validation data used in the early stopping. The total number of leaves per tree and the minimum number of training points per terminal leaf were both set to $N^{0.5}$, where $N$ is the number of training points in a given cluster. The number of leaves per tree determines the size of the tree. The difference with *LightGBM* compared to other packages, like *XGBoost*, is that trees are grown on a leaf wise basis rather than a level-wise basis, where the depth of the tree would be a more important hyper-parameter. The minimum number of training points per terminal leaf determines how many points are aggregated in an estimate – a small number could thus result in over-fitting. The value $N^{0.5}$ was determined experimentally with a single ensemble member, where the optimal values were determined

with K-Fold cross validation. The results were in the ball-park of $N^{0.5}$ showing relatively low sensitivity to changes in these hyper-parameters. Further, the learning rate was set to 0.2 and $L1$ and $L2$ regularization were both set to 20.

### A3.3   $p\text{CO}_2$: Feed-forward neural network

Given that the problem of solving $p\text{CO}_2$ is not very complex (*i.e.* it is within the capability of a single layer neural network), the multi-layer perceptron regressor from the *scikit-learn* package was used. The size of the hidden layer for each cluster was determined by shuffled K-fold cross validation with five splits. The maximum number of weights in a hidden layer was set to $N^{0.55}$. Back propagation was performed using the Adam optimizer. The learning rate of the optimizer was selected in the cross validation process. Early stopping was used to speed up the training process and prevent over-fitting where one random third of the data were used in early stopping.

### A4   GRaCER mapped cluster metrics

One of the advantages of using the GRaCER approach is that any metric can be mapped from the results to the appropriate clusters, resulting in an ensemble of metric scores. The possible metrics that can be applied include bias, root mean squared error, and mean absolute error. Further, these metrics can be applied to test data, meaning that the resulting scores can be based on test scores — that is data that is unseen by the model during the training process, thus giving a true representation of the uncertainty. Given that the cluster used in this study are climatological, we can get fully mapped climatological estimates of uncertainty.

The uncertainty of TA remains fairly constant between summer and winter, with the Amazon plume showing increased uncertainty in northern hemisphere summer (Figure A2c).

The seasonal difference is larger for $p\text{CO}_2$ than for TA. For example, uncertainty in the Southern Ocean is much larger in the southern hemisphere summer (Figure A2b, DJF) compared with winter (Figure A2d, JJA).

Similarly, the spatial distribution of feature importances can be determined with the GRaCER approach when using Gradient Boosted Decision Trees as the regression method (Figure A3). Each ensemble member has a feature importance assigned to a cluster. When averaged over the ensemble members, a smoothed climatological estimate of feature importance can be estimated.

### A4.1   Cluster boundaries

The GRaCER method introduces the idea of using an ensemble of clusters, thus removing the variability that may be introduced in the clustering step. The location of the clusters varies from ensemble member to ensemble member. This creates a "high variance — low bias" scenario that is used by other ensemble methods such as Random Forests (Breiman, 2001). The location of these boundaries can give information about the mean distribution of the clusters. For example, locations where there are no cluster boundaries ($\leq 1$ in Figure A5) indicate cluster centers that fall within the same cluster for the majority of time steps

and ensemble members. While regions where cluster boundaries occur very often (> 8 in Figure A5) are indicative of regions where boundaries are found for most time steps and ensemble members.

*Author contributions.* LG & NG conceived of the study and developed the method. LG performed the analysis and testing of the method and wrote the paper with substantial input by NG.

*Competing interests.* Both authors declare that they don't have any competing interests.

*Acknowledgements.* We are deeply indebted to the scientists who sampled, analyzed, and contributed to the global data bases for ocean carbon data, namely the Surface Ocean $CO_2$ Atlas (SOCAT) and the Global Ocean Analysis Project (GLODAP). We also thank the funding agencies that made these efforts possible. SOCAT and GLODAP are international efforts, endorsed by the International Ocean Carbon Coordination Project (IOCCP), the Surface Ocean Lower Atmosphere Study (SOLAS) and the Integrated Marine Biosphere Research (IM-BeR) program. This work was financially supported by ESA's OceanSODA project (contract No. 4000112091/14/I-LG) and the European Commission through Horizon 2020 research and innovation programme under grant agreements 821003 (4C) and 820989 (COMFORT). We thank Meike Vogt, Fabio Benedetti, and Peter Land for their valuable input and discussions. We are indebted to Jamie Shutler for initiating and leading the OceanSODA project and for his substantial input to our quantification of the uncertainty budget.

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

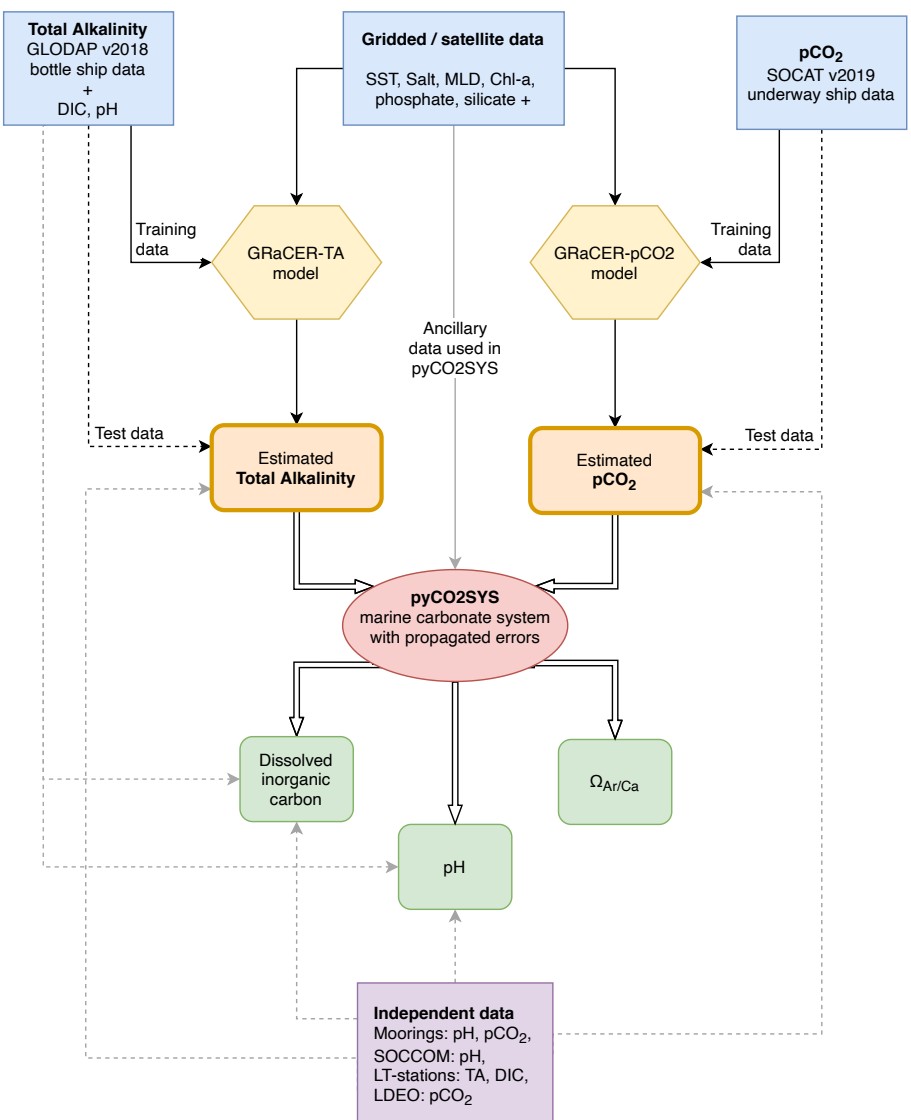

**Figure 1.** Schematic flow diagram showing the three steps required to reconstruct the the surface ocean carbonate system. In the first step (yellow hexagons), the GRaCER (Geospatial Random Cluster Ensemble regression) method is used to develop statistical models for the observed TA (left) and $p$CO$_2$ (right) fields. In the second step (orange rectangles), these statistical models are used to extrapolate these two parameters over time and space using ancillary observations, primarily stemming from satellite observations. In the third step (red oval), the inter- and extrapolated TA and $p$CO$_2$ fields are then used to compute the remaining parameters of the surface ocean carbonate system, namely $DIC$, pH, and the saturation state of seawater with regard to mineral CaCO$_3$, $\Omega$. The output of steps two and three is the OceanSODA-ETHZ product. Also shown are the various data sets and data flows used in this study. The different lines indicate whether data is used for training (solid lines), testing (dashed lines), or output with an estimate of uncertainty, where independent test data are shown with gray dashed lines. The gridded/satellite data are summarized in Table 1. Independent test data are shown by the purple box. *pyCO2SYS* is the software used to solve the marine carbonate system and propagate uncertainties (Humphreys et al., 2020).

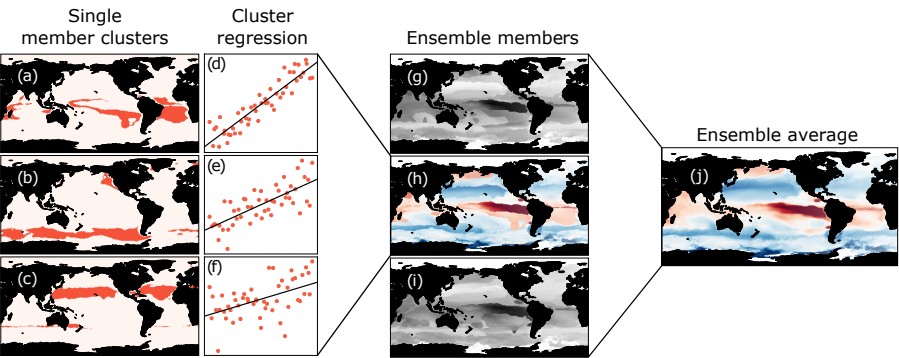

**Figure 2.** A schematic showing the steps used in the GRaCE-R method for a single month. (a-c) show a subset of the clusters of a single ensemble member (h), with the adjacent scatter plots (d-f) showing the training data for each cluster and the linear regression models for that cluster (with toy data). (g-i) show the ensemble member estimates for a subset of three members for $pCO_2$. (j) shows the ensemble mean for all ensemble members, which includes ensemble members not shown in (g-i).

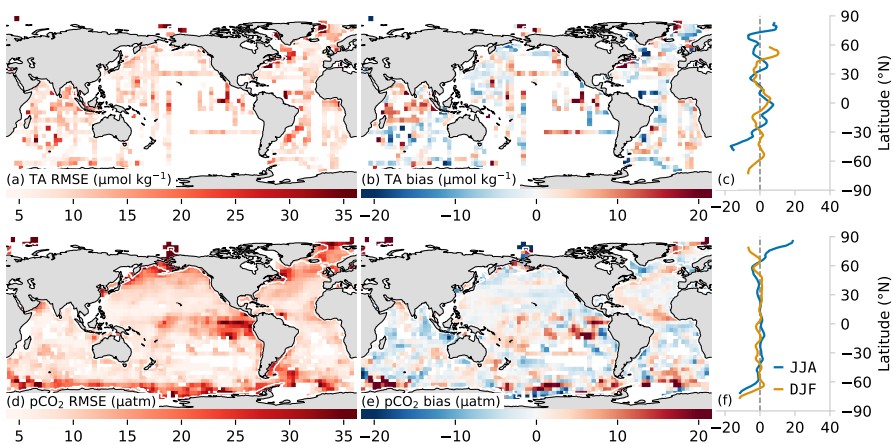

**Figure 3.** Test metrics for total alkalinity (a,b) and $pCO_2$ (c,d). The left-hand column (a, d) shows the root mean squared error (RMSE) compared with the target data. Similarly, the middle column (b, e) shows bias compared to the respective training data sets (GLODAP v2.2019 and SOCAT v2019). The right-hand column shows the zonally-averaged biases for June, July and August (JJA, blue), and December, January and February (DJF, orange). A 2D spatial convolution was first applied to the $1° \times 1°$ pixels to make regional patterns in the biases and RMSE clearer and data were then aggregated into $4° \times 4°$ pixels for clearer visualization.

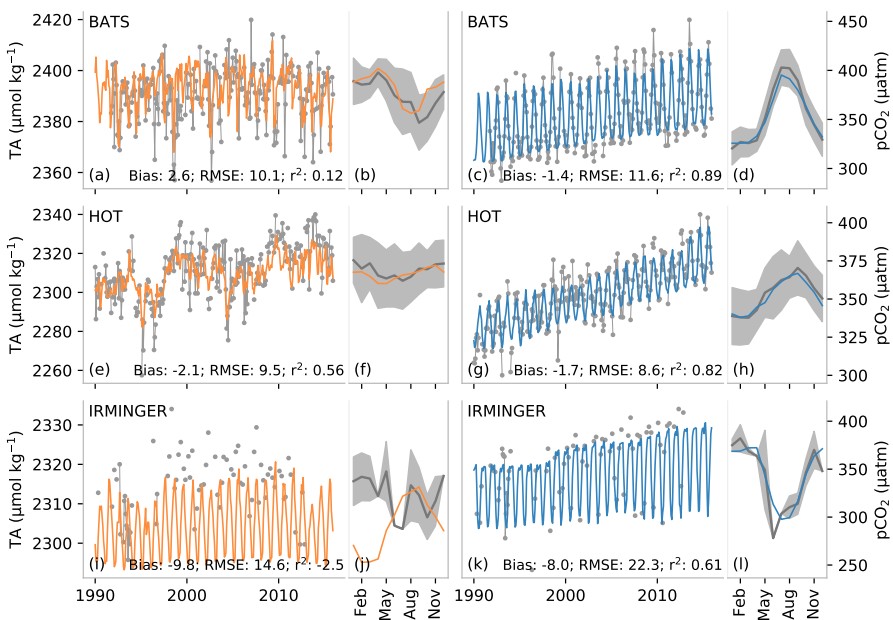

**Figure 4.** A comparison of a subset of measurements from long term observation stations (gray) with predicted total alkalinity (TA) (left: *a,b,e,f,i,j*) and partial pressure of $CO_2$ (*p*$CO_2$) (right: *c,d,g,h,k,l*). The top row (a-d) shows data for the Bermuda Ocean Time Series (BATS), the middle row (e-h) for the Hawaii Ocean Time-series (HOT), and the bottom row (i, l) shows the Irminger station. The narrow panels show the average of the seasonal climatology for the time series. The gray shading shows the standard deviation of the observations for the period 1990 to 2018, while the orange/blue lines show the average estimate. TA for the Irminger station is calculated from *p*$CO_2$ and DIC, and *p*$CO_2$ is calculated for BATS and HOT using DIC and TA, as described in section 2.4.

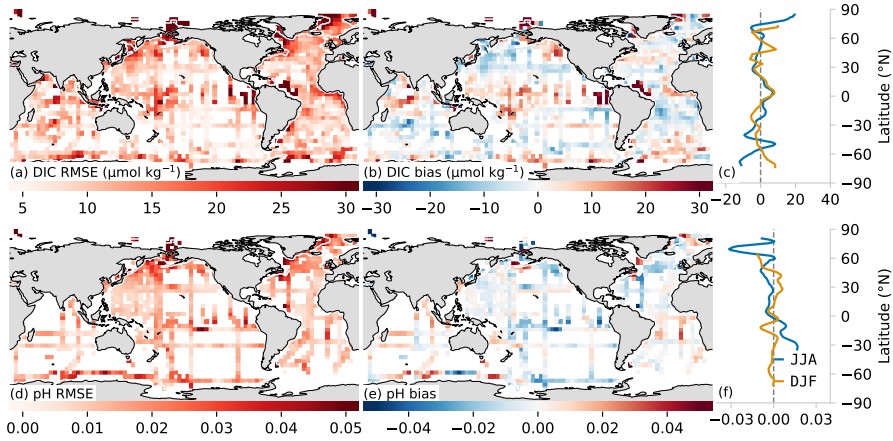

**Figure 5.** Root mean squared error (RMSE - a,d) and biases (b,c,e,f) for: dissolved inorganic carbon (DIC, top) and pH (bottom) compared with in situ GLODAP v2.2019 data. The two subplots in the right most column compare the zonally-averaged bias for JJA and DJF. Data were processed for plotting as described in Section 4.

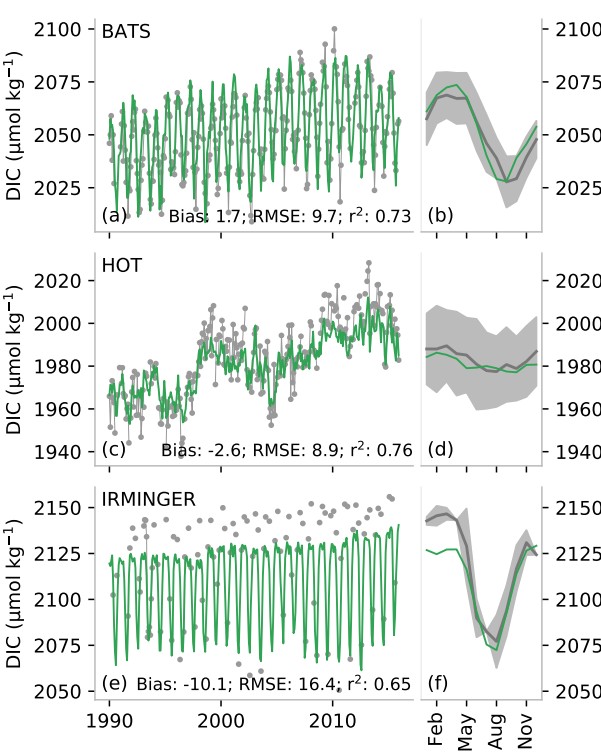

**Figure 6.** A comparison of observations from long term observation stations (gray) with predicted dissolved inorganic carbon (DIC). The top row (a,b) shows data for the Bermuda Ocean Time Series (BATS), the middle row (c,d) for the Hawaii Ocean Time-series (HOT) and the bottom row (e,f) shows the Irminger station. The narrow panels on the right show the average of the seasonal climatology for the time series, where the gray shading shows the standard deviation of the observations for the period 1990 to 2018, while the green line shows the average estimate.

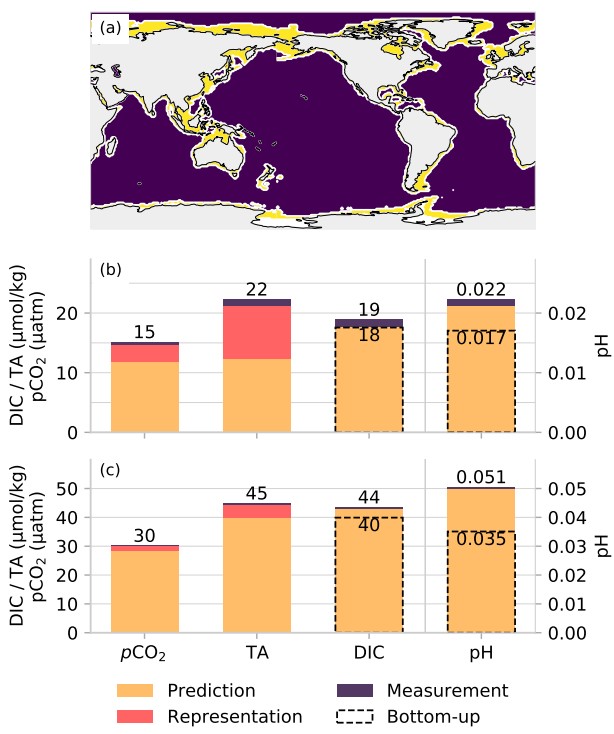

**Figure 7.** A comparison of propagated uncertainties with independent uncertainties as an assertion of the validity of uncertainty estimates. The map (a) shows the separation between coastal and open ocean, (b) shows the error contributions in the open ocean, and (c) in the coastal ocean. The total uncertainty has been broken into the three different components. Note that the values represented by the bar plots are not equivalent to values in Table 3 as the latter shows $p\mathrm{CO_2}$ and TA total uncertainties for test data only, while the bar charts show total uncertainties for all data; further the breakdown of the error contributions is proportional to the contribution of the sum of the squares (see Eq. 2).

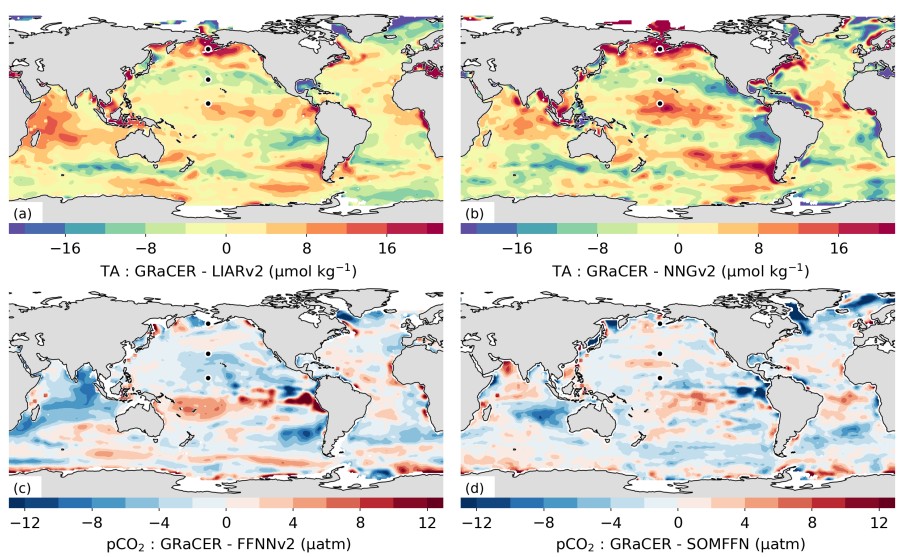

**Figure 8.** A comparison of the mean differences between TA (top) and $pCO_2$ (bottom) for OceanSODA-ETHZ and other published methods: (a) LIARv2, (b) NNGv2, (c) CMEMS-FFNNv2, and (d) MPI-SOMFFN. The markers in the North Pacific show the locations used in the comparison of the climatology in Figure 11.

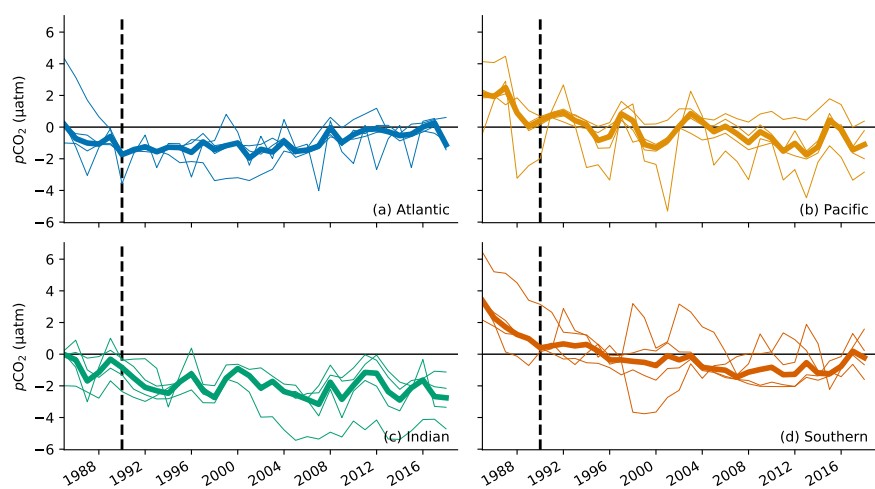

**Figure 9.** A basin-mean comparison of OceanSODA-ETHZ $pCO_2$ with four gap-filling methods: MPI-SOMFFN, JENA-MLS, CMEMS-FFNNv2, and CSIR-ML6. The thin lines show the differences to the individual methods, while the thick line shows the mean difference across the four methods. We do not show the Arctic Ocean as OceanSODA-ETHZ covers only 23% of the region. The vertical dashed line in each figure marks the year 1990, where estimates prior to this period show biases in (b, c).

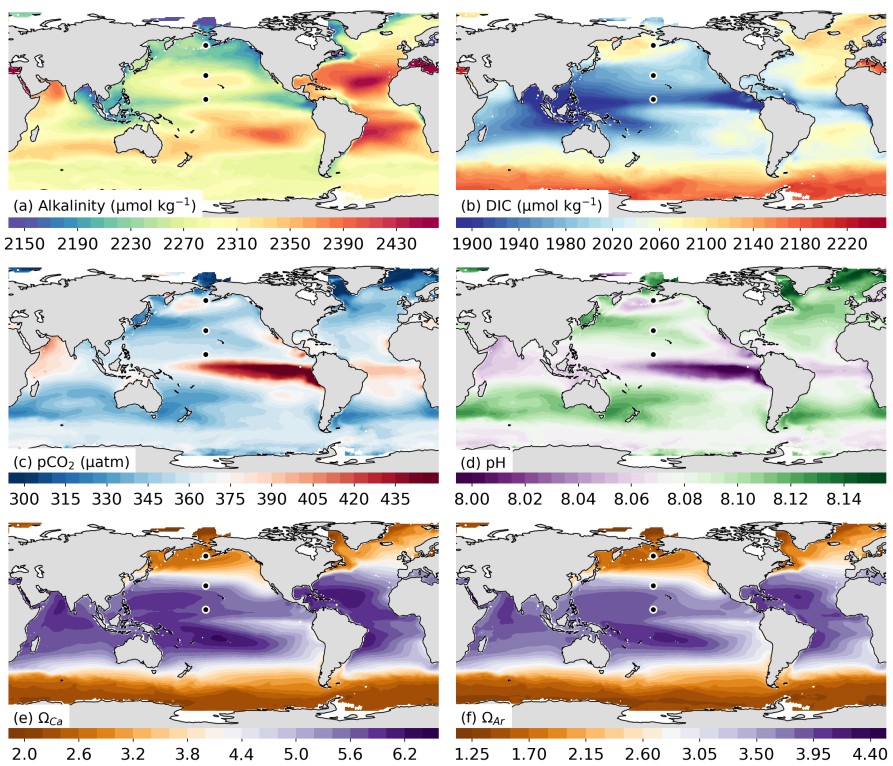

**Figure 10.** Mean maps of the GRaCER-based estimates for the period 1985–2018 for (a) total alkalinity and (c) $p\text{CO}_2$, as well as those of the computed variables, (b) dissolved inorganic carbon, (d) pH, (e) $\Omega_{calc}$ (saturation state with regard to calcite), and (f) $\Omega_{arag}$ (saturation state with regard to aragonite). The three black markers in each plot show the locations chosen for the seasonal analysis in Figure 11*f-j*.

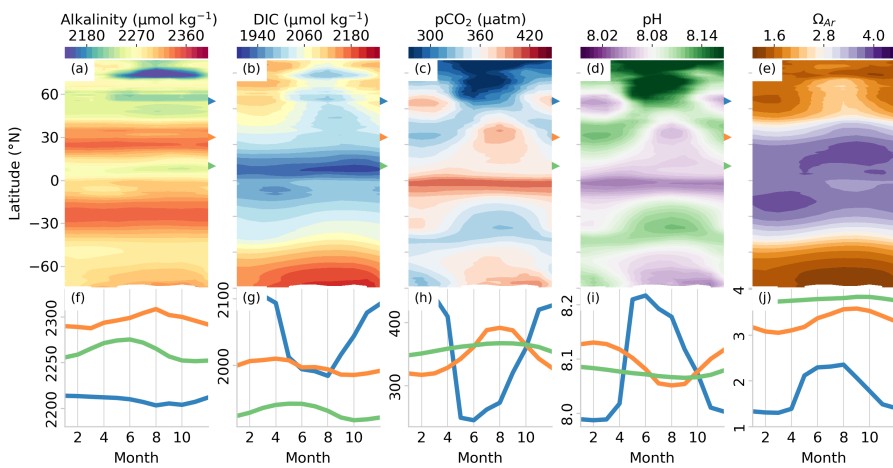

**Figure 11.** Hovmoeller plots (*a-e*) showing the zonally averaged seasonal climatologies for (a) total alkalinity, (b) dissolved inorganic carbon, (c) $p\mathrm{CO_2}$, (d) pH, and (e) aragonite saturation state ($\Omega_{Ar}$). The second row of panels (*f-j*) show the corresponding variables for a high (55°N, 180°E, blue), mid (30°N, 180°E, orange) and low-latitude (10°N, 180°E, green) location. The units for (*f-j*) correspond with the units in (*a-e*).

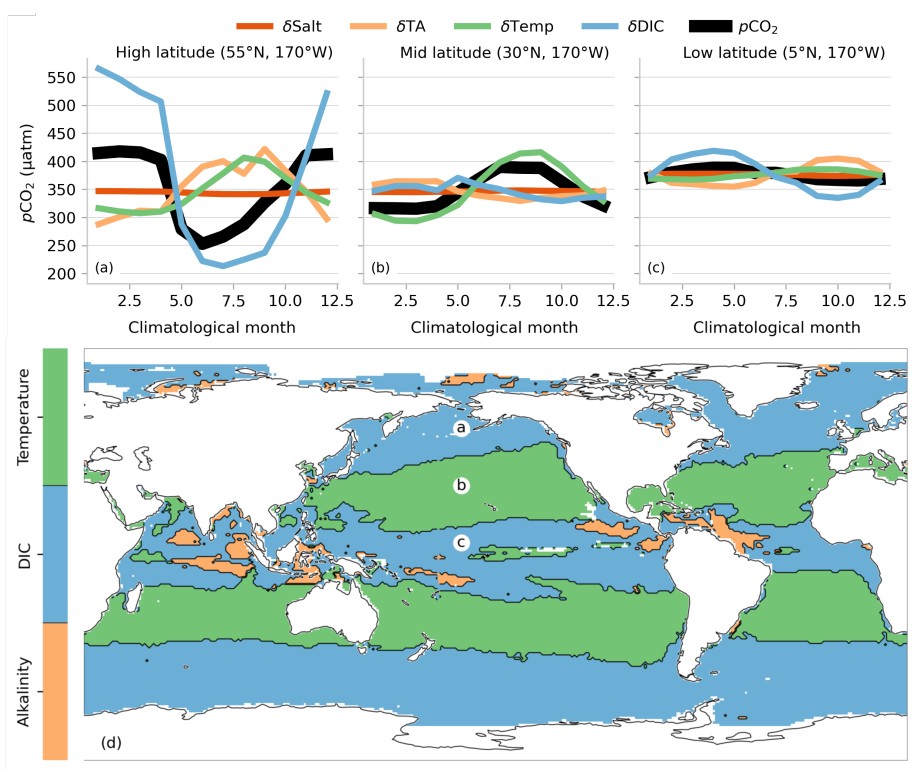

**Figure 12.** Attribution of DIC, TA and temperature to the seasonal cycle of $pCO_2$. The top two figures show the seasonal cycle of $pCO_2$ and the drivers thereof for (a) high latitudes, (b) mid latitudes, and (c) low latitudes. These locations are shown with the markers in the Pacific ocean in (c). The map (d) shows the dominant driver of the seasonal cycle for each region calculated as the value with the maximum seasonal amplitude.

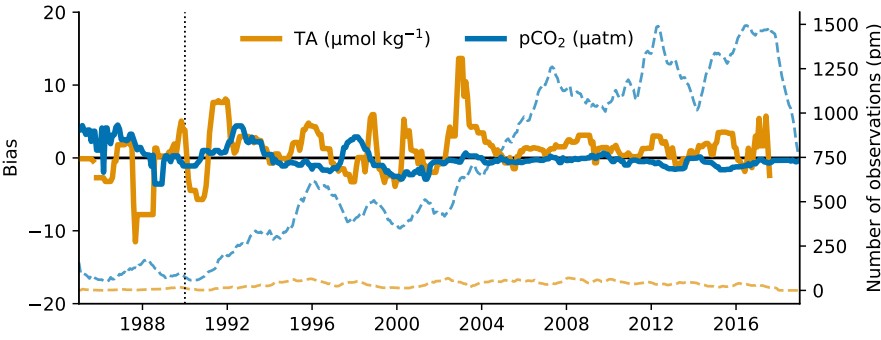

**Figure A1.** Time series of TA (orange) and $pCO_2$ (blue) estimates, with the respective training data sets (GLODAPv2 and SOCAT). The dashed lines show the number of training data (right axis). The vertical line shows the year 1990, before which OceanSODA-ETHZ $pCO_2$ estimates tend to be larger than other gap-filling estimates of $pCO_2$

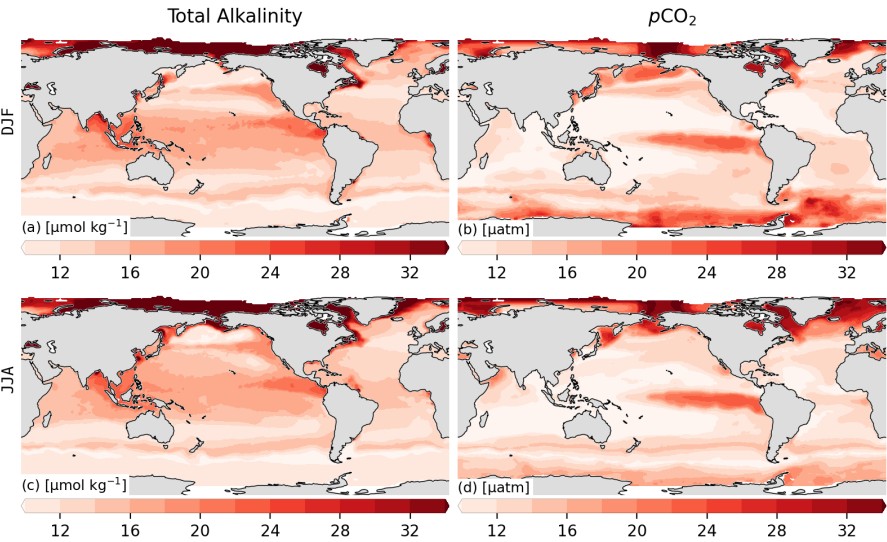

**Figure A2.** The Huber test scores mapped to the ensemble clusters for Total Alkalinity (TA) and $pCO_2$. The top row shows Huber scores averaged for December, January, and February (DJF) and the bottom row June, July, and August (JJA). The Huber score is a blend between root mean squared error (RMSE) and mean absolute error (MAE), where MAE is applied to values that are considered outliers. Only test data is used to calculate these climatological scores, meaning that the scores are based on GLODAP2 and SOCAT data for TA and $pCO_2$ respectively.

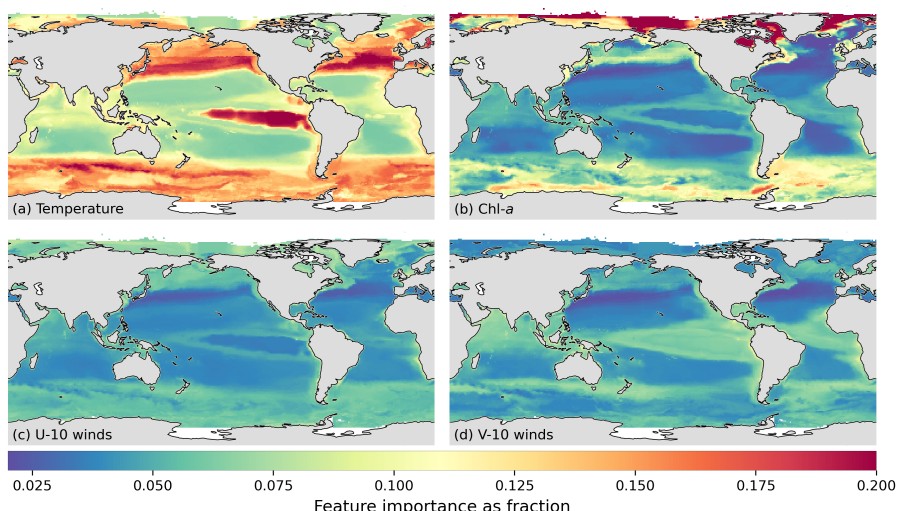

**Figure A3.** Feature importances determined by Gradient Boosted Decision Trees for $pCO_2$ predictions. A subset of four proxies are shown for the months of June, July, and August. The feature importances allow one to make informed decisions about the inclusion or exclusion of proxy variables. Here, temperature (a) is one of the more important features, Chl-$a$ is most important in the high northern latitudes, the $U$ and $V$ components of the winds are important along the coastal regions, particularly the eastern boundary upwelling systems.

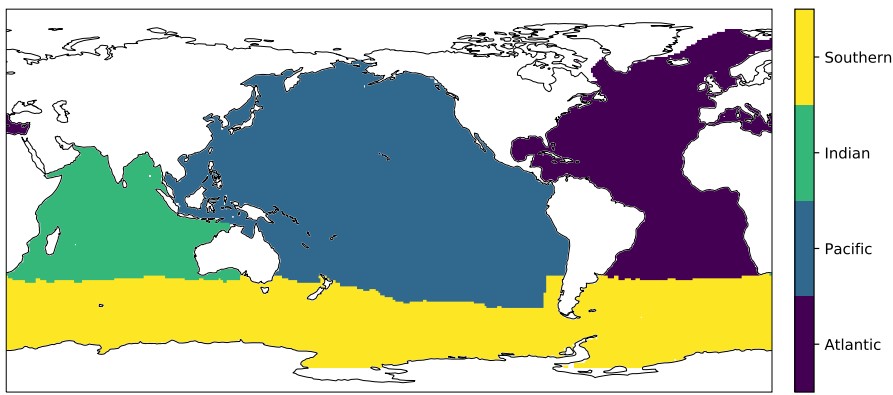

**Figure A4.** Ocean basin boundaries used in Table 5 as used by the RECCAP2 project (https://reccap2-ocean.github.io/regions/). The Southern Ocean and North Atlantic boundaries are defined by biome boundaries defined in Fay and McKinley (2014).

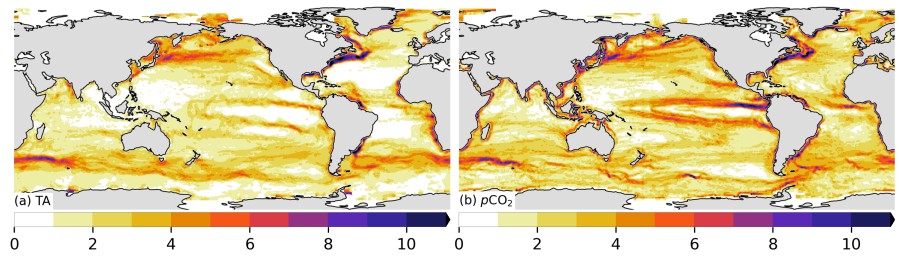

**Figure A5.** Map of the position of cluster boundaries across all ensemble members and months for (a) total alkalinity and (b) $p\mathrm{CO_2}$. The white regions indicate locations that belong almost exclusively to the same cluster. Dark regions show where cluster boundaries are persistent.