# Peer review of "OceanSODA-ETHZ: A global gridded data set of the surface ocean carbonate system for seasonal to decadal studies of ocean acidification"

_Earth System Science Data, 2020_

## Referee Comment (RC1) · Anonymous Referee #1 · 17 Nov 2020

**Review of Gregor and Gruber et al. (2020)**

**General Comments:**

This manuscript introduces a new global dataset for carbonate chemistry parameters (DIC, TA, $pCO_2$, pH, and omega for $CaCO_3$) over 1985–2018. The development and production of this dataset was clearly a significant effort, and should prove useful to the broad ocean biogeochemistry community. In general, I found the paper to be logically structured / readable and the figures are of good quality. The section on uncertainty assessment was a useful contribution and most welcome. However, some revisions are required to improve the overall quality of the text and discussion section of the paper.

**Specific Comments:**

Discussion: it would be good to add 1–2 paragraphs on potential issues in the coastal zone compared to open ocean and recommendations for future efforts. It would also be useful to connect to this recent paper: Landschützer, P., Laruelle, G. G., Roobaert, A., and Regnier, P.: A uniform pCO2 climatology combining open and coastal oceans, Earth Syst. Sci. Data, 12, 2537–2553, https://doi.org/10.5194/essd-12-2537-2020, 2020.

Discussion: you mention ocean models briefly in the introduction (L45). It would be helpful for the community if you could make some recommendations (based on what you have learned in the development of this paper) for accurately simulating and benchmarking ocean acidification in numerical ocean models.

Appendix A3.4 seems to be missing, please correct this.

**Technical corrections / comments:**

General: omega symbol and "omega" are used interchangeably, please stick to one throughout the text to be more consistent and readable.

General: "In situ" is italicized in some sections and not in others. Please make this consistent.

Table 4: Suggest shading "this study" rows as light gray to set it apart from the other methods.

Figure 1: Would be helpful to add a label for steps 1–3.
Figure 4: Suggest making the gray scatters and shading darker, it is difficult to see.
Figure 5 (caption): Add hyphen between "zonally" and "averaged" (same in Figure 10). Add "Section" before "4".
Figure 9 (caption): Need coma after "calcite)".
Figure 10: Need labels for colors shown in *f-j*.
Figure 11: Can you move location labels for top figures above the plots? Especially in (a) where the label overlaps the blue line.
Figure 1A: Need comma after "July".

L1: Consider moving "profoundly" to before "altered".

L2: Add comma after "long-term"; suggest rewording from "that permits to study" to "allows for the study".

L4: Add hyphen between "methodologically" and "consistent".

L10: Suggest replacing "higher" with "improved".

L24: Suggest changing "lowering" to "moderating" and changing "But" to "However,".

L25: Need space between "acidification" and "(Doney …)".

L26: Suggest removing "the" before "global mean".

L28: Remove "the" before "H+".

L29: Add "ocean" between "surface" and "$pCO_2$".

L30: Suggest changing to: "due to the ~1-year timescale".

L32: Would be good to add a reference for the decadal atmospheric $CO_2$ increase timescale.

L34: Add "the" before "mineral".

L36: Would be good to define "calcium carbonate" on L34, then use abbreviation of $CaCO_3$ in rest of text.

L39: Suggest rewording to "cross critical saturation thresholds".

L44: Add comma before "respectively".

L50: Suggest changing to: "local-to-regional".

L54: Remove "among others".

L62: Change "And" to "Additionally,".

L63: Change "slaved to" to "dependent on".

L64: Add hyphen between "well" and "measurable".

L65: Add comma after "pH".

L76: Add comma after "(2013)".

L77: Suggest rewording to: "all surface ocean carbon system parameters".

L78: Add comma before "due".

L84: Change "accessibility to" to "accessibility of".

L85: Need parenthesis before "Olsen".

L88: Remove comma after "historically".

L89: Change to "have permitted a tremendous increase in the number of pH" and remove "the" before "deployments" and "biogeochemical Argo".

L96: Add "us" before "to address" and change "But" to "However,".

L108: Change "But" to "However," and change ":" to ";".

L109: Add hyphen between "data" and "sparse".

L113: Add hyphen between "highly" and "linear".

L115: Change to: "a globally-application, multi-linear".

L121: Change to: "5th".

L124: Add hyphen between "regionally" and "specific".

L126: Remove "only".

L127: Remove "concretely". Suggestion joining sentences with "map TA; however, their application".

L128: Remove "then".

L133: Suggest changing to: "map TA and $pCO_2$ globally," and removing "will" before "use a newly". Also add hyphen between "newly" and "develop" and "two" and "step".

L137: Add hyphen between "methodologically" and "consistent".

L149: Suggest rewording to: "First, we develop a statistical".

L153: Change "to the globe" to "globally".

L170: Change "permits to overcome" to "overcomes".

L181: Change hyphen "-" to em dash "—". Same on L206.

L183: Add hyphen between "computationally" and "expensive".

L196: Suggest adding "($NO_3$)" after nitrate and then using "$NO_3$" and "$PO_4$" instead of "nitrate" and "phosphate" on L206 (and throughout the rest of the text too). These are shown in Table 2, but the reader doesn't see that until Section 2.3. Also, the name and abbreviation for silicic acid are used interchangeably in the text, please make that consistent.

L212: Change "But" to "However," and add comma after "Arctic" and "layer depth" on L213 and L214, respectively.

L232: Need parenthesis around block of references, i.e., right after $pCO_2$.

L240: Add hyphen between "temporally" and "averaged".

L249: Add comma after "so".

L253: Remove extra "the".

L259: Can you explain and justify why you take the log(10) of Chl-a here?

L294: Note that you have degree symbols after both 1's here. This is not consistent throughout the text, please adjust accordingly.

L301: Add space between "2.5" and "μatm".

L311: Add comma after "phosphate".

L326: Add hyphen between "statistically" and "modeled".

L328: Remove "established".

L334: Suggest showing equation for E with sqrt symbol and not $E^2$.

L349: Add hyphen between "closely" and "related".

 Table 3 caption: need closing parenthesis for "(see".

L361: Add hyphen between "overly" and "conservative".

L362: Need space between "5" and "μmol".

L367: "In-situ" does not need to be capitalized; add hyphen between "satellite" and "based".

L385: Change hyphen to en dash in "1990-2018", also L386 add comma after "HOT -2 μmol $kg^{-1}$"

L394: Missing period at end of sentence.

L401: Add space after "0.5x0.5" and "x 15-days". Also, add degree symbols to be consistent with rest of text.

L428: "/kg" should be "$kg^{-1}$" to be consistent with rest of text.

L439: Again, "/kg" should be "$kg^{-1}$" to be consistent with rest of text. Also, on L465.

L480: Need comma after "pH".

L500: Subscript "2" in "[CO2]". Also, add "Figure" before "10c,g)".

L520: Need degree symbols instead of "deg".

L522: Change hyphen to em dash.

L538: Suggest changing "not so" to "not as".

L570: Add hyphen between "monthly" and "gridded".

L595: Change "estimates" to "estimate".

L611: Need to subscript "2" in "pCO2". Also, on L638, 640, and 641 (for "CO2").

L651: Need comma after "γ".

L663: Suggest changing "further" to "furthermore".

---

## Referee Comment (RC2) · Anonymous Referee #2 · 19 Nov 2020

**General Comments:**

The manuscript introduces a complete and novel carbonate chemistry observation-based dataset, that will be useful to detect changes in the carbonate system. The $pCO_2$ dataset compares well to previous similar datasets, but the extrapolation method used here incorporates new and improved features, and the alkalinity observation-based dataset is novel. The $pCO_2$ and TA can be used together to calculate other important parameters such as DIC and pH.

The GRaCER method is similar to previously used techniques, but introduces an ensemble of cluster-regressions that solve the boundary problem between different regions. The authors do an excellent job at calculating possible dataset errors and biases and do an extensive analysis of it. The method and the resulting dataset are an important contribution to the scientific community. However, I believe there are some improvements that need to be made to the manuscript before being accepted for publication. The authors need to address some possible issues with the clustering method, and whether the dataset is suitable for analysis on time-scales longer than the seasonal cycle. Below I detail these points and some technical corrections.

**Specific Comments:**

**Methods section:**

I believe the novelty of the GRaCER method is that produces an ensemble of clusters. The ensemble is produced because the clustering process randomly assigns the first cluster center in the predictor-variable space. Thus, each member of the ensemble has a different center, and therefore the ensemble mean does not have discrete boundaries. Based on this, it would be helpful to clarify a couple of details:

- Lines 172-179: How does the result vary, if you use monthly data instead of climatologies for the clustering process?

- Lines 245-248: "It may seem tautological to use other machine learning estimates, but these data are just used to create regional clusters, i.e., they are not used in the regression step. " How do the results vary if you do not use the previous machine learning estimates for clustering? Previous methods (Landschutzer, Rodenbeck etc), only use the observations from SOCAT for clustering. In this manuscript these datasets are not used in the regression step, but I believe that the results are affected by which data is used for the clustering.

- For the non-expert reader it would be helpful to know how is the "bias" measured. This word is repeated through the manuscript but is not clear how it is calculated. I suggest a short explanation in the appendix.

**Discussion section:**

The title of the paper indicates that this dataset can be used for studies from seasonal to decadal periods, but the authors do not analyze or compare the trend and the inter-annual and decadal characteristics of this dataset with respect to SOCAT or other observation-based datasets.

A section should be added in the discussion to address whether the dataset is suitable for studies on time-scales longer than seasonal, to answer "How well represented is the inter annual and decadal variability?". Similar observation-based datasets may not be suitable for interannual variability; for example the Rodenbeck dataset indicates that "*interannual variations* may miss important features" (see http://www.bgc-jena.mpg.de/CarboScope/?ID=oc "period of validity" and "usage notes"). Also, see for example Gallego et al., 2019 in which is discussed the lower interannual variability of the $pCO_2$ dataset of Landschutzer et al. 2018, when compared to the CMIP5 models. In a pre-print Gloege et al., 2020 show that the neural-network estimates are not as skillful for inter annual variability as for seasonal cycle estimations (see https://www.essoar.org/doi/10.1002/essoar.10502036.1).

If the authors consider that their dataset is good to estimate internal and decadal variability, then they should add some analysis and comparison with other existent $pCO_2$ observation-based datasets.

Moreover, some methodologies used in the manuscript should be discussed in the context of inter annual to decadal variability, such as:

- The MLD sub-annual and inter annual variability is removed.

- To calculate DIC and pH they use climatologies of silicic acts and phosphate instead of monthly data.

- A $pCO_2$ RMSE of 12 muatm is larger than inter annual variability for many locations.

- Lines 412-415: "The highest biases on $pCO_2$ when comparing with observations are located in the eastern tropical Pacific where inter annual variability is higher". This may suggest that the dataset does not represent well inter annual variability.

- The cluster seems to collect data by climatological month. How does he method change if monthly data is used instead?

**Results section:**

Figure 8 (d) shows the mean of the monthly differences between the SOMFFN and GRaCER pCO2 datasets. When I plot the time-series of the monthly differences between SOMFFN and GRaCER averaged over the eastern equatorial Pacific, I see a continuous decrease in the difference from 1985-2018. That could mean that for the beginning of the time period there is a larger difference between the two methods than by the end. Could this be a consequence of using the SOMFFN and other products for the clustering process? Since at the beginning of the period there is less observational points compared to the end.

[Figure]

Monthly difference averaged over the eastern eq. Pacific

[Figure]

**Technical corrections / comments:**

**Figure 2:** Which data is shown in these plots? This is an interesting illustrative figure, but in line 172 of the paper it says "we use monthly climatological data of $pCO_2$ and TA and related parameters (Figure 2a-c)…" But the caption on Figure 2 does not specify which data (TA or pCO2) is shown, and from (d) to (f) which data are used to do the regression.

**Line 176:** The cluster center seems to be what determines that clusters and therefore it is possible to create an ensemble of them. However, it is not clear for the non-expert reader what is "the cluster center" or how it is defined. Is it one arbitrary data point?

**Line 256:** The removing of sub-annual and inter annual MLD variability, seem that it could cause an underestimation of the inter annual variability of $pCO_2$.

**Figure 3:** Where it says (a,c) should say (a,d), and where it says (b,d) should say (b,e). Also in the caption it would be helpful for the reader to indicate what are panels (c,f).

**Line 387:** I suggest to explain further the sentence "Further, the mean seasonal cycle well is relatively well represented at HOT and BATS, being within one standard deviation of the interannual variability (Figure 4b,f). ".
- It seems to me that the mean climatology cannot be one standard deviation from the inter annual variability.

**Figure 7 caption:** Suggest "further the breakdown of the errors is proportional to the contribution of the sum of the squares (see Eq. (2))"**.**

**Figure 9 caption:** Perhaps would be helpful to also indicate in the caption that the maps show the 1985-2018 mean, since the word climatology would suggest we are looking at a monthly means.

**Line 40 and 50:** Show "Omega" instead of $\Omega$.

**Line 143:** At the end there is an extra ")"

**Caption Figure 1, 7th line:** Should say "our" instead of or?

**Lines 231-232:** References need to be inside brackets.

**Line 253:** It says "the the"

**Line 343:** instead of "$1x1^o$ by month" should say "$1^o x1^o$ monthly grid"

**Lines 346-347:** I suggest a rewording of ".And with an order of magnitude fewer observations of TA, an even larger number of cells are populated by a single observation. "

**Caption Figure 4:** Change for "The top row (a-d) shows data for the Bermuda Ocean Time Series (BATS), the middle row (e-h) for the Hawaii Ocean Time-series (HOT), and the bottom row (i, l) shows the Irminger station."

**Line 387:** Remove "well" from "the mean seasonal cycle well is relatively well".

**Caption Table 4:** It would be good to clarify which product is LDEO (Takahashi).

**Line 456:** Correct the sentence "However, the estimates are not as coherent for pH where there the bottom-up error is 23% smaller top-down error than the in the open ocean. "

**Line 466:** Suggest changing "in the spatial distribution of the distribution, " to "in the spatial pattern of the distribution"

**Caption Figure 10:** Hovmoeller plots (*a-d*) should say (a-e).

**Line 493:** Change "update" for "uptake"

**Line 498:** instead of 10,c,d,g,h should be 10,c,d,h,i

**Line 500:** Should be $CO_2$ instead of CO2 and 10c,g should be 10,g,h

**Line 507:** Suggest to change to: "This would also be true for Omega_calc which only differs from Omega_arag in …"

**Line 546:** Suggest to change w.r.t. for the meaning (with respect to?) since the abbreviation seems unnecessary.

**Line 553:** For the read it would help to clarify what it means the representation error of the horizontal representation errors.

**Figure 11 caption:** Last line should say The map (d).

**Line 579:** I suggest adding the reference of Gallego et al. 2018 (Drivers of future seasonal cycle changes in oceanic $pCO_2$ ) and Fassbender et al., 2018 (Seasonal Asymmetry in the Evolution of Surface Ocean $pCO_2$ and pH Thermodynamic Drivers and the Influence on Sea-Air $CO_2$ Flux).

**Section A3.4:** Seems to be empty.

**Line 655:** Remove one "every"

---

## Author Comment (AC1) · 24 Dec 2020

**Response to Reviewer 1 (R1)**

We thank reviewer 1 for their prompt and positive feedback. Their review was detailed and was a great help in preparing the revised manuscript. Below, we outline how we will address the major points raised : 1) recommendation on how modelers should use the data set; 2) discussion of coastal estimates , particularly considering the new merged Landschützer et al. (2020) product; 3) missing appendix. We included below the reviewer's comments in blue and our responses in black. Italic green font indicates text that will be added to the manuscript. The location of the text in the manuscript is indicated if applicable. The line specific grammar issues will be addressed and shown in the track-changed document. Note that we have also refined our definition of error and uncertainties. These changes have also been marked in the tack changes file.

**1. Inclusion of the coastal ocean**

*It would be good to add 1–2 paragraphs on potential issues in the coastal zone compared to open ocean and recommendations for future efforts. It would also be useful to connect to this recent paper: Landschützer, P., Laruelle, G. G., Roobaert, A., and Regnier, P.: A uniform pCO2 climatology combining open and coastal oceans, Earth Syst. Sci. Data, 12, 2537–2553, https://doi.org/10.5194/essd-12-2537-2020, 2020.*

We will add a paragraph in the discussion about the validity of our coastal estimates. In particular, we will outline why we have some confidence in these estimates (with the figure below supporting our point), even though there also some clear limits. Further we will make reference to the MPI-ULB-SOMFFN merged product. Lastly, we will emphasize that users should consult the original SOCAT data for comparison if they would like to use the product beyond the climatological scale. We plan to include the following text in the discussion.

[Figure]

***Caption (Appendix):*** *Map of the position of cluster boundaries across all ensemble members and months for (a) total alkalinity and (b) pCO2. The white regions indicate locations that belong almost exclusively to the same cluster. Dark regions show where cluster boundaries are persistent.*

***Discussion:*** *The OceanSODA-ETHZ product extends further into the coastal margin than most previous studies (Iida et al., 2015; Land-schützer et al., 2016; Denvil-Sommer et al., 2019). This is achieved i) by including coastal observations during the training, and ii) by using a larger number of clusters compared to other clustering approaches (Landschützer et al., 2016; Watson et al., 2020). This permits to better separate open ocean and coastal variability through the inclusion of suitable variables in the clustering step (e.g. Chl-a for $pCO_2$, and see Figure A3 to see a representation of cluster boundaries). This gives us some*

*confidence in the coastal estimates, at least on a climatological scale with regard to the seasonal cycle. Our product is therefore comparable to that of Landschützer et al. (2020) who blended separate coastal and open ocean pCO₂ products into a single climatological product with monthly resolution (Landschützer et al., 2016; Laruelle et al., 2017).*

*The total uncertainties of our estimates in the coastal ocean are considerably larger compared to the open ocean estimates (Figure 7). This reflects the much higher spatio-temporal variability of the physical and chemical environment in the coastal ocean, leading to much higher variations in the marine carbonate system (Laruelle et al., 2017). Since our predictor variables are only partially reflecting this variability, a large portion of the high total uncertainty is due to a high representation error (Table 3). Increasing the resolution of the products may improved coastal estimates as done by Laruelle et al. (2017). Until we arrive at this point, the OceanSODA-ETHZ data should be used with care in the coastal ocean. Further, we recommend that researchers interested in the investigation of interannual variability and trends in the coastal ocean using the OceanSODA-ETHZ product should also look a the underlying in situ data to gain a better understanding of the variability, trends, and uncertainties for the coastal region of interest.*

**2. Recommendation to modelers**

*You mention ocean models briefly in the introduction (L45). It would be helpful for the community if you could make some recommendations (based on what you have learned in the development of this paper) for accurately simulating and benchmarking ocean acidification in numerical ocean models.*

We will add a section with specific recommendations to users of the product, containing a paragraph aimed at the modeling community. Specifically, we will add an additional uncertainty estimate that provides climatologically mapped errors based on test data that has not been seen by the trained model. This will permit modelers to assess model-data misfits on a local basis, thereby making model-data comparisons more quantitative.

[Figure]

***Figure A2:** The Huber test scores mapped to the ensemble clusters for Total Alkalinity (TA) and $pCO_2$. The top row shows Huber scores averaged for December, January, and February (DJF) and the bottom row June, July, and August (JJA). The Huber score is a blend between root mean squared error (RMSE) and mean absolute error (MAE), where MAE is applied to values that are considered outliers. Only test data is used to calculate these climatological scores, meaning that the scores are based on GLODAP2 and SOCAT data for TA and $pCO_2$ respectively.*

***Discussion:** The OceanSODA-ETHZ product provides a useful comparison for numerical models with the full marine carbonate system. The spatially resolved climatological estimates of uncertainty for TA and pCO2, based on in situ data, provide useful context for ocean modelers on a climatological time-frame (Figure A2). In the same way that previous studies have used pCO2, the OceanSODA-ETHZ data set can also be used to compare interannual trends and variability of the marine carbonate system (Landschützer et al., 2015, 2016; Gregor et al., 2018; Keppler and Landschützer, 2019).*

*(**Appendix**) One of the advantages of using the GRaCER approach is that any metric can be mapped from the results to the appropriate clusters, resulting in an ensemble of metric scores. The possible metrics that can be applied include bias, root mean squared error, and mean absolute error. Further, these metrics can be applied to test data, meaning that the resulting scores can be based on test scores — that is data that are unseen by the model during the training process, thus giving a true representation of the uncertainty. Given that the cluster used in this study are climatological, we can get fully mapped climatological estimates of uncertainty.*

Missing appendix

*Appendix A3.4 seems to be missing, please correct this*

This will be corrected to include text on how the variable importance figure was created. The main text now will refer to the figure instead of the section in the appendix.

---

## Author Comment (AC2) · 24 Dec 2020

**Response to Reviewer 2 (R2)**

We thank reviewer 2 for their prompt and positive feedback. R2 provided an in-depth critique of the method and of the data product itself. The major concerns raised include: 1) The use of a gap-filled pCO2 climatology to perform clustering; 2) the use of climatological nutrients in solving the marine carbonate system; 3) uncertainties of the dataset, particularly over time. We have included the reviewer's comments in italicized blue font, while our answers are given in black. We grouped all comments and answers into these three topics as there is significant overlap in many of the reviewer's comments. We have also refined our definition of error and uncertainties. These changes have also been marked in the tack changes file.

**1. Use of gap-filled pCO2 to cluster**

*I believe the novelty of the GRaCER method is [it] that produces an ensemble of clusters. The ensemble is produced because the clustering process randomly assigns the first cluster center in the predictor-variable space. Thus, each member of the ensemble has a different center, and therefore the ensemble mean does not have discrete boundaries. Based on this, it would be helpful to clarify a couple of details:*

- *Lines 245-248: "It may seem tautological to use other machine learning estimates, but these data are just used to create regional clusters, i.e., they are not used in the regression step." How do the results vary if you do not use the previous machine learning estimates for clustering? Previous methods (Landschutzer, Rodenbeck etc), only use the observations from SOCAT for clustering. In this manuscript these datasets are not used in the regression step, but I believe that the results are affected by which data is used for the clustering.*
- *Lines 172-179: How does the result vary, if you use monthly data instead of climatologies for the clustering process?*
- *The cluster seems to collect data by climatological month. How does he method change if monthly data is used instead?*

The reviewer raised a question about our using of pCO2 climatologies (called pCO2map) that are based on similar machine-learning methods. This indeed could come across as tautological, especially since two of the products we are using were derived using related methods, i.e., the the SOM-FFN estimate of Landschützer et al. (2013) and the LSCE-FFNN estimate of Denvil-Sommer et al. (2019). We consider this issue as very minor, since use these climatologies just for the clustering step, and not for the final estimation of pCO2. This is done through our regression approach. Extensive testing has shown that our results are very robust with regard to the choice of pCO2map. This is because, all four mapping methods are able to predict the seasonal cycle of pCO2 well (Sommer-Denvil et al. 2018), which is the main dimension of variability that is captured by the clustering step.

Thus one could argue that given this lack of sensitivity, one should restrict pCO2map to just the truly independent methods, such as the LDEO climatology. We feel that the benefit of using multiple products in pCO2map outweighs this alternative. The use of an ensemble avoids overfitting in regions of where the LDEO climatological distribution is noisy (see

Figure). It also permits us to cluster regions that are not covered by the LDEO climatology. The following change was made to the manuscript with underlined text being inserted.

*Clustering is performed on climatological values of pCO2, SST, mixed layer depth and Chlorophyll-a, with additional weighting given to pCO2. As with TA, all variables are standardized prior to clustering with (x - μ) / σ, after which pCO2 is multiplied by 3 to give it stronger weighting. The larger weight given to pCO2 means that monthly clustering would result in very similar results to climatological clustering, as only SST and Chl-a would vary over time and not pCO2. Details of the regression method, and of the hyper-parameter selection are given in section.*

[Figure]

**Will not be added to manuscript:** Owing to the substantial amount of ""noise" present in the LDEO climatology stemming from the way the measurements are interpolated (b) it is prone to create spurious clusters. Further, the coverage of pCO2map (a) allows for clustering and thus predictions in regions not mapped by the LDEO climatology, such as the Mediterranean.

We also prefer to stick with our original choice of using the climatological distribution of pCO2 (and Alk) for the clustering. First, as also pointed out by the reviewer, there is much less confidence in the interannual (monthly) estimates. Using these data for clustering rather than the climatology of these products would run the risk of creating problems given that such a step would also further increase the weight of the products. This is because pCO2 is weighted three times more than the other variables in the clustering step. Second, and from a more fundamental perspective, we argue that such a step would undermine a strength of the two step approach, i.e., its separation of variations on different timescales. The clustering step is meant to isolate primarily regions with the same seasonal cycle. The regression step is meant to explain the variability within each region. This is based on the assumption that, to first order, interannual variability can be considered as modifications of the seasonal cycle. If we were to allow the clusters to vary inter annually, we would lose this fundamental distinction, and we would ask the SOM step to take over a bigger burden of the total variance. Given the discrete nature of the mapping, this can actually lead to worse results.

In summary, we consider our choice to be well justified. At the same time, our experience indicates that it is unlikely that the outcome of the regression is impacted much by the details of the clustering step. We added some small comment on this issue to the method section. The inserted text is underlined:

*The main advantage of such a two-step approach is that the first clustering step organizes the variability regionally and temporally. This greatly enhances then the fidelity of the second step, i.e., the regression, as the size of the regression problem is reduced from the global*

*domain to smaller, more homogeneous regions. A second advantage is that this clustering brings together regions with similar seasonality and similar co-variability with potential predictors, irrespective of the number of observations. The regression step explains the variability within each region over time and space dimensions, including interannual variability. Further, the clustering permits the regression to transfer information from spatially distant, but geochemically similar regions, making the inter and extrapolation more robust in data poor regions.*

*...*

*For the clustering step, we use monthly climatological data of pCO2 and TA and related parameters (Figure 2a-c), to deter-mine the main patterns of variability of the target variable and its co-variability with potential predictor variables. Concretely, the clustering step is meant to isolate primarily regions with the same seasonal cycle.*

*Figure 8 (d) shows the mean of the monthly differences between the SOMFFN and GRaCER pCO2 datasets. When I plot the time-series of the monthly differences between SOMFFN and GRaCER averaged over the eastern equatorial Pacific, I see a continuous decrease in the difference from 1985-2018. That could mean that for the beginning of the time period there is a larger difference between the two methods than by the end. Could this be a consequence of using the SOMFFN and other products for the clustering process? Since at the beginning of the period there is less observational points compared to the end.*

No, various tests showed that this trend in difference is not a consequence of our use of SOMFFN estimats in our clustering. The first part of the difference is largely due to the fact that data are very sparse. This makes the mapped estimates more sensitive to the specifics of the methods, especially in the regression step (not in the clustering step). In the regression step, SOMFFN and GRaCER are actually quite different, which explains the divergence of the estimates. In response, we emphasize in our new section on data use that the first part of the timeseries should be used with great caution.

*The MLD sub-annual and inter annual variability is removed.*

We use an observationally based mixed-layer depth product that is only available as a climatology (Holte et al. 2018). This product reports the mixed layer depth for Argo profiles globally and is not normalized to a specific year. Hence, the interannual variability of the MLD is still present in this data product. In a climatological context, this interannual variability acts as noise rather than signal. We thus remove this interannual signal using a Gaussian smoother.

**2. Climatological nutrients for marine carbonate system calculations**

*To calculate DIC and pH they use climatologies of silicic acts and phosphate instead of monthly data.*

Using climatological concentrations of $PO_4$ and $SiO_4$ in the calculations of the marine carbonate system instead of the interannually varying concentrations has a very small impact on the computed values. We base this conclusion on the following worst-case scenario, i.e., that the year to year variability is larger than the seasonal cycle in these nutrients. To illustrate this, we take a location in the Southern Ocean (60S, 40W) characterized by a very large seasonal cycle, and vary silicic acid ($59 \pm 15$ µmol/kg) and phosphate ($1.7 \pm 0.3$

µmol/kg) over this seasonal range.  As shown by the figure below, the maximal impact of using climatological nutrients in this calculation is 1.8 µmol/kg. In reality, the range of interannual variations will be much smaller. Thus, we consider the potential implication of our using climatological nutrient concentrations instead of interannually varying ones as neglibile. In response, we will add to the text  that this assumption has very little impact.

[Figure]

**Not added to manuscript:** The range of DIC when using a range of phosphate (PO4) and silicic acid (Si) to solve the marine carbonate system from pCO2 and TA. The input ranges for PO4 and Si were determined from the magnitude of the seasonal cycle from a region where the variability is large.

**3.  Uncertainty related points**

*Moreover, some methodologies used in the manuscript should be discussed in the context of interannual to decadal variability*

We will add a section in the discussion of the manuscript that gives recommendations for the use of the OceanSODA-ETHZ data (as per request of R1). In this section we caution users of the product to treat data prior to 1990 with care as sparse data results in substantial uncertainties in the estimates, and also larger differences  between the different methods as shown in Watson et al. (2020). This will be accompanied by the figure shown below.

***Discussion – Recommendations for use:*** *However, users of the OceanSODA-ETHZ product should be aware of the fact that that data prior to the 1990's should be treated with care due to the paucity of SOCAT pCO2 training data during this period (Rödenbeck et al., 2015; Watson et al., 2020). This was rescently demonstrated by Watson et al. (2020) who used an ensemble of various regression approaches to show that the spread of pCO2 estimates prior to the 1990's is large due to the paucity of data. Similarly, Gregor et al. (2019) showed that pCO2 estimates prior to 1990 tend to have aslightly positive bias. Hence, the trends shown in Table 5 are calculated for the years after 1990, i.e., covering the period 1990–2018.*

*If the authors consider that their dataset is good to estimate internal and decadal variability, then they should add some analysis and comparison with other existent pCO2 observation-based datasets.*

Agreed. In response, we will add the figure below to the manuscript showing the basin-mean difference between OceanSODA-ETHZ pCO2 and that of the other methods: MPI-SOMFFN, Jena-MLS, CMEMS-FFNN, and CSIR-ML6. The thin lines show the individual method differences, while the thick line shows the mean difference of the four methods. In the Indian ocean, the OceanSODA-ETHZ pCO2 estimate is persistently lower than the pCO2 estimated by the ensemble of the four other methods. A similar negative difference is found in the Atlantic, but the difference diminishes from 2008 onward. In the Pacific and Southern Ocean, there are positive differences prior to 1990 that diminish thereafter. The spread of the relative differences is larger in the Pacific and Southern Ocean, not unexpected given the much larger data gaps in these ocean basins.

[Figure]

*Results – Comparison with other products: basin-mean difference between OceanSODA-ETHZ pCO2 and other methods: MPI-SOMFFN, Jena-MLS, CMEMS-FFNN, and CSIR-ML6. The thin lines show the differences to the individual methods, while the thick line shows the mean difference across the four methods.*

*Results – Comparison with other products: We also show the basin-mean temporal differences between OceanSODA-ETHZ pCO2 and other gap-filling methods (Figure 9). In the Atlantic (Figure 9a), OceanSODA-ETHZ pCO2 is < 2 µatm lower than the mean of the other gap-filling methods for the period 1990 to 2008. Thereafter, the difference is < 1 µatm. In the Indian ocean, our pCO2 estimates have a persistent negative difference of ~ 2 µatm (Figure 9c). The comparison in the Pacific (Figure 9b) is the most consistent with the other methods, with a slight positive difference in the beginning of the period (pre-1990). The OceanSODA-ETHZ estimates of pCO2 in the Southern Ocean (Figure 9d) have a large positive difference prior to 1990 – up to 6 µatm for one of the ensemble members. This difference quickly diminishes and is near zero by 1990. There is also a negative difference later in the period (2004 to 2015); however, the ensemble spread over this period is large.*

*The comparison with other methods illustrates that while gap-filling methods are converging on a global scale, there are regionally differences. Further, large differences in pCO₂ between methods prior to 1990 indicates high uncertainty for this period.*

To further address R2's point, we will add the figure of the bias of the TA and pCO2 relative to the training data sets (GLODAPv2 and SOCAT respectively) in the supplementary material. The biases for $pCO_2$ are larger (~5 µatm) at the beginning of the time series when there is less data. Gregor et al. (2019) found similar biases for $pCO_2$ in the pre-1990 period using gradient boosted trees. TA biases are erratic over time due to the highly uneven sampling distribution (*e.g.*, not only in time, but also in space, e.g., by sampling in river plume areas where the uncertainty is large). The low number of samples exacerbates this effect.

[Figure]

***Appendix:*** *Timeseries of the median bias (solid lines) of TA (orange) and pCO₂ (blue) relative to the GLODAP and SOCAT datasets, respectively. The dashed lines show the number of observations for each of the data (right axis).*

*A pCO₂ RMSE of 12 muatm is larger than inter annual variability for many locations.*

The RMSE represents a distribution of the uncertainty theoretically centered around a zero mean. It provides an estimate of the expected uncertainty of a particular instance. In our case, this is an estimate for a single grid cell for a single month. When assessing variations in pCO2 (or any of the other quantities), we usually analyze the temporal variations averaged over a larger region or averaged over an entire year. Even though the data are to a certain degree autocorrelated in time and space which reduces the number of degrees of freedom somewhat, the uncertainty of the mean estimate is reduced by the square-root of the (reduced) number of degrees for freedom. For example, when averages are formed over 10°x10° regions in the annual mean, we expect at least a factor of ten reduction in the uncertainty of this mean. Thus, we expect an uncertainty of around 1µatm of this mean, which is much smaller than the signal one is interested.

When looking at long-term trends, potential biases matter as well. The biases in our product are in general ≪ 10 µatm, with the exception of parts of the Southern Ocean, and the Eastern Tropical Pacific (see paragraph below). Fortunately, there are well sampled areas (through time) in these regions allowing us to compare the OceanSODA-ETHZ $pCO_2$ estimates with SOCAT directly. In the figure below we show direct comparisons of $pCO_2$ in the Eastern Equatorial Pacific, Western Pacific, and the Drake Passage. The interannual variability is well captured in these regions, confirming our hypothesis. Still, the large interannual variability in the Eastern Equatorial Pacific is occasionally underestimated.

[Figure]

**Not added to manuscript:** Comparison of OceanSODA-ETHZ (solid) and SOCAT (dashed) pCO2 for open ocean regions. The three chosen regions have persistent occupancy for the selected regions and periods: 62% for the Eastern Equatorial Pacific (< 5°N/S < 180°W), 84% for the Western Pacific (25° to 40° N, 128° to 145°E), and 76% for the Drake Passage (> 50°S, between 73° and 65°W over the period 2000 to 2018).

*Lines 412-415: "The highest biases on $pCO_2$ when comparing with observations are located in the eastern tropical Pacific where inter annual variability is higher". This may suggest that the dataset does not represent well inter annual variability*

The analyses above demonstrate that the Ocean-SODA-ETHZ product is able to capture most of the large variability observed in the Eastern Tropical Pacific. The large RMSE and biases are found at the edges of the tropical Pacific, where the mean lateral gradients are high. These lateral gradients shift strongly during El Niños and La Niñas, posing a challenge to any interpolation method. Any small deviation in the specific location of the gradient leads to large local biases, although the large-scale spatial mean is well captured. The fact that the interannual variability of the Eastern Equatorial Pacific is still well captured (as shown above) reinforces this.

**4. Extra additions**

Lastly, we have added a table showing the trends of a set number of variables of the marine carbonate system. The trends are calculated for the period 1990-2018 due to the reasons explained above.

***Results – Regional Trends:*** *The global and basin-scale trends for $pCO_2$ are remarkably consistent ($\sim 16.5$ µatm decade$^{-1}$) when compared with $pCO_2^{atm}$ ($\sim 18.6$ µatm decade$^{-1}$), with the atmospheric slope being slightly steeper than the oceanic trend (Table 5). The basin-scale consistency holds true for pH (-0.016 units decade$^{-1}$) and $\Omega_{ar}$ (-0.07 units decade$^{-1}$), where global values are consistent with the regional values. Total alkalinity trends are more variable from basin to basin but are driven almost entirely by salinity with a basin-scale correlation of 0.99. Lastly, DIC shows similar variability to TA trends, which makes sense in terms of TA increasing the buffering capacity of seawater while $pCO_2$ remains relatively consistent on a basin-scale.*

***Results – Regional Trends:*** *Table showing the slopes and associated standard error for OceanSODA-ETHZ variables for the period 1990-2018. All columns show increases per decade (d). All trends are significant (P > 0.05). We exclude the Arctic as the OceanSODA-ETHZ product only covers 23% of this region and may thus give spurious trends. The Ocean basins are defined by the map shown in Figure A4.*

| Region | pH | $\Omega_{AR}$ | TA | DIC | $pCO_2$ | $pCO_2^{atm}$ |
|---|---|---|---|---|---|---|
| Units | (units/d) | (units/d) | (µmol/kg/d) | (µmol/kg/d) | (µatm/d) | (µatm/d) |
| Global | -0.015 ± 0.0 | -0.06 ± 0.0 | 1.4 ± 0.1 | 8.2 ± 0.1 | 15.5 ± 0.1 | 17.8 ± 0.1 |
| Atlantic | -0.015 ± 0.0 | -0.06 ± 0.0 | 3.5 ± 0.1 | 9.9 ± 0.3 | 15.7 ± 0.1 | 17.9 ± 0.1 |
| Pacific | -0.015 ± 0.0 | -0.07 ± 0.0 | 0.6 ± 0.1 | 7.9 ± 0.2 | 15.7 ± 0.1 | 17.8 ± 0.1 |
| Indian | -0.015 ± 0.0 | -0.07 ± 0.0 | 3.0 ± 0.3 | 9.7 ± 0.4 | 15.3 ± 0.3 | 17.6 ± 0.1 |
| Southern | -0.016 ± 0.0 | -0.06 ± 0.0 | 0.2 ± 0.1 | 6.7 ± 0.5 | 15.0 ± 0.3 | 17.7 ± 0.1 |

[Figure]

***Figure A4:*** *Ocean basin boundaries used in Table (above) as used by the RECCAP2 project (https://reccap2-ocean.github.io/regions/). The Southern Ocean and North Atlantic boundaries are defined by biome boundaries defined in Fay and McKinley (2014).*

---

## Author Response (AR2)

**Response to topical editor's comments**

Dear Dr Manzella,

Thank you for your feedback. We have made the changes requested. Primarily, we have changed the references of Olsen et al. (2016) to the (2019) version. The other changes are shown in the track changes document.